# DeepVideo-R1: Video Reinforcement Fine-Tuning via Difficulty-aware Regressive GRPO

**Jinyoung Park**[1][*]    **Jeehye Na**[1][*]    **Jinyoung Kim**[2]    **Hyunwoo J. Kim**[1][†]

[1]Korea Advanced Institute of Science and Technology,    [2]Korea University

{jinyoung.park, jeehyena, hyunwoojkim}@kaist.ac.kr
k012100@korea.ac.kr

https://github.com/mlvlab/DeepVideoR1

## Abstract

Recent works have demonstrated the effectiveness of reinforcement learning (RL)-based post-training for enhancing the reasoning capabilities of large language models (LLMs). In particular, Group Relative Policy Optimization (GRPO) has shown impressive success using a PPO-style reinforcement learning algorithm with group-normalized rewards. However, the effectiveness of GRPO in Video Large Language Models (VideoLLMs) remains underexplored. In this paper, we explore GRPO and identify two issues that hinder effective learning: (1) reliance on safeguards, and (2) vanishing advantage. To mitigate these challenges, we propose **DeepVideo-R1**, a video large language model trained with **Reg-GRPO** (**Reg**ressive **GRPO**) and difficulty-aware data augmentation. Reg-GRPO reformulates the GRPO loss function as a regression task that directly predicts the advantage in GRPO, eliminating the need for safeguards such as clipping and min operations. This directly aligns the model with the advantages, providing guidance to prefer better outputs. The difficulty-aware data augmentation strategy augments input prompts/videos to target solvable difficulty levels, enabling diverse reward signals. Our experimental results show that our approach significantly improves video reasoning performance across multiple benchmarks.

## 1 Introduction

Large Language Models (LLMs) [1–4] have demonstrated remarkable abilities in understanding, reasoning, and generating text across diverse domains. Their success stems from next-token prediction over vast corpora, which enables the emergence of complex reasoning patterns and world knowledge. Building on this progress, recent research has extended LLMs into the video domain, giving rise to Video Large Language Models (VideoLLMs) [5–9]. These models aim to unify video understanding and language generation, enabling capabilities such as temporal event reasoning, video question answering, and video-to-text summarization.

Despite their rapid evolution, current VideoLLMs still struggle with complex reasoning tasks, which require temporal, spatial, and semantic understanding over video sequences. Since standard supervised fine-tuning fits instruction data rather than reasoning processes, it is limited in improving reasoning capabilities. To address this, reinforcement learning (RL)-based post-training [10, 11] has emerged as a compelling paradigm. RL provides a mechanism to optimize models beyond likelihood objectives, aligning them with reward signals that encode human preference or task-specific success. Recently, Group Relative Policy Optimization (GRPO) [12, 13] has shown promise by using group-based advantages and relative preference signals to enhance reasoning capabilities.

---

[*]Work was done at Korea University.

[†]corresponding author.

39th Conference on Neural Information Processing Systems (NeurIPS 2025).

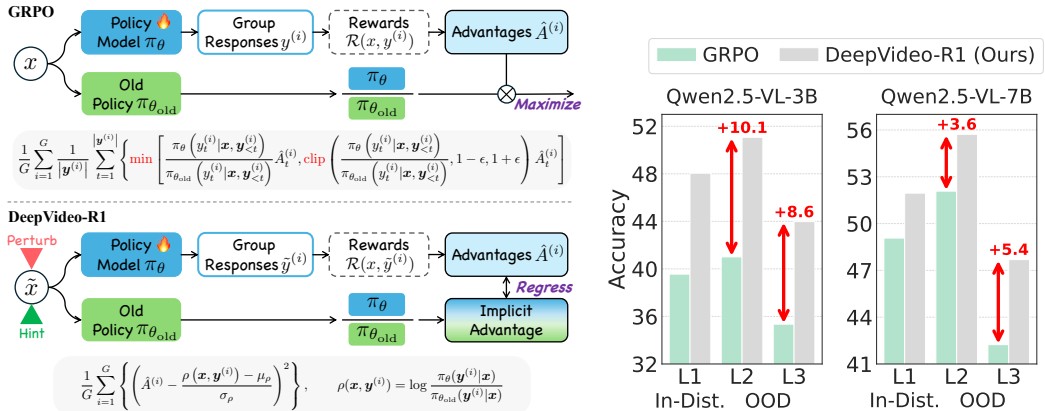

Figure 1: **DeepVideo-R1 significantly improves the reasoning capabilities of VideoLLMs.** Our VideoLLM, DeepVideo-R1, is trained to explicitly predict the advantage $\hat{A}^{(i)}$ through Regressive GRPO loss. Notably, model training becomes significantly effective and achieves a 10.1 performance improvement compared to GRPO.

While GRPO has achieved strong results in text-based tasks, its application to VideoLLMs remains underexplored. In this work, we investigate the application of GRPO to VideoLLMs and identify two key limitations that hinder effective training: (1) reliance on stabilizers such as minimum and clipping operations, which often suppress gradients and impede convergence, and (2) the vanishing advantage problem, where extremely easy or difficult samples yield zero advantages, thereby removing the training signal.

To overcome these limitations, we propose **DeepVideo-R1**, a video large language model trained with two key innovations: *Regressive GRPO* (Reg-GRPO) and *difficulty-aware data augmentation*. Reg-GRPO reformulates the GRPO objective as a regression problem that directly predicts group-based advantage values. This simple yet effective reformulation enables direct alignment between model outputs and the advantage values, eliminating the need for stabilizers while ensuring stable convergence. We also introduce a difficulty-aware augmentation that dynamically adjusts the difficulty of video-text inputs. For easy samples, we perturb the video content to inject uncertainty; for hard samples, we provide auxiliary reasoning cues. This strategy diversifies the reward landscape, mitigating the vanishing advantage problem and promoting balanced learning across difficulty levels.

Our experimental results demonstrate the effectiveness of DeepVideo-R1 on multiple challenging video reasoning benchmarks such as SEED-Bench-R1, LongVideoBench, and NExTGQA, achieving superior performance over recent video LLMs such as Qwen2.5-VL [7] (Figure 1). Notably, our model achieves consistent improvements on both in-distribution and out-of-distribution tasks, indicating robust generalization capabilities. These results underscore the benefits of combining a regression-based RL objective with data augmentation for training large-scale multimodal reasoning models.

Our main contributions are listed as:

- We introduce **Reg-GRPO**, a novel optimization scheme that casts GRPO as a regression task over group-based advantage values, eliminating heuristic stabilizers such as clipping and min operations, and mitigating the vanishing gradient issue.

- We develop a **difficulty-aware augmentation** framework that modulates video-text inputs with adaptive difficulty scaling, video cue injection, and noise perturbation to generate richer and more effective reward signals.

- We propose **DeepVideo-R1**, a video large language model trained with two key innovations: Regressive GRPO (Reg-GRPO) and difficulty-aware data augmentation. Our experimental results demonstrate that DeepVideo-R1 significantly improves the reasoning capabilities of VideoLLMs on complex video reasoning tasks.

## 2 Related Work

**Video Large Language Models (VideoLLMs).** Large Language Models (LLMs) [14–16] have exhibited strong generalization and reasoning capabilities across a wide range of domains, including knowledge-intensive tasks [17–20], mathematical reasoning [12, 21], and scientific domains [22–24]. Building on these capabilities, Video Large Language Models (VideoLLMs) have extended LLM reasoning to dynamic video domains, achieving notable performance across tasks [25–28], such as video question answering [7–9, 29–31] and video captioning [32–34]. Despite their impressive performance, VideoLLMs remain limited on long video inputs and fine-grained video understanding tasks that require detailed spatiotemporal reasoning [35–37]. Most existing methods primarily emphasize video perception or short-context understanding, often relying on static supervised fine-tuning objectives that fail to capture reasoning dynamics. To address these challenges, we leverage a reinforcement learning-based fine-tuning approach to improve the reasoning and generalization capabilities of VideoLLMs.

**RL-based fine-tuning.** Multiple works [12, 13, 38–41] have significantly improved the reasoning capabilities of LLMs through reinforcement learning (RL), including DPO [42] and RLHF [43]. Recently, variants of RL-based fine-tuning [44, 45] have explored direct reward-regression losses derived from RL objectives. A key development in this direction is Group Relative Policy Optimization (GRPO), an RL algorithm proposed in [12] that computes group-wise normalized rewards to stabilize training and improve efficiency. Motivated by GRPO, several approaches have demonstrated substantial improvements in the reasoning abilities of MLLMs across image [46–53] and video tasks [54–59]. While existing approaches [55, 57, 56] have primarily focused on defining appropriate reward functions tailored to each visual task, some concurrent works [51, 52] instead study practical issues that arise during GRPO training, aiming to further enhance model reasoning. In this work, we propose a learning algorithm that directly regresses advantages instead of increasing the likelihood of high-advantage responses. Additionally, we design difficulty-aware data augmentation to provide diverse and dense learning signals.

## 3 Methods

In this section, we present a video large language model named DeepVideo-R1, which is trained with Regressive GRPO (Reg-GRPO) and difficulty-aware data augmentation for effective video context reasoning. We first introduce post-training methods for VideoLLMs, such as proximal policy optimization and group-relative policy optimization (GRPO), and discuss the limitations of GRPO: *reliance on heuristic safeguards* and *vanishing advantage*. Then, we propose Reg-GRPO, which improves the RL-based GRPO by transforming the objective into a simpler yet more effective regression loss, eliminating heuristic safeguards such as the min and clipping operations. Finally, we present a novel difficulty-aware data augmentation, which alleviates the vanishing advantage problem by modulating the difficulty of samples.

### 3.1 RL-based Fine-Tuning

**Proximal Policy Optimization (PPO) [60]** is one of the most widely used actor-critic RL algorithms for fine-tuning (video) large language models. For example, RLHF [61] applies the PPO algorithm. Given the input sample $\boldsymbol{x}$, PPO optimizes the model $\pi_\theta$ with the following objective:

$$\mathcal{L}_{\text{PPO}}(\theta) = -\mathbb{E}_{\boldsymbol{x}, \, \boldsymbol{y} \sim \pi_{\theta_{\text{old}}}(\cdot|\boldsymbol{x})}$$
$$\frac{1}{|\boldsymbol{y}|} \sum_{t=1}^{|\boldsymbol{y}|} \min \left[ \frac{\pi_\theta(y_t|\boldsymbol{x}, \boldsymbol{y}_{<t})}{\pi_{\theta_{\text{old}}}(y_t|\boldsymbol{x}, \boldsymbol{y}_{<t})} A_t, \text{clip}\left( \frac{\pi_\theta(y_t|\boldsymbol{x}, \boldsymbol{y}_{<t})}{\pi_{\theta_{\text{old}}}(y_t|\boldsymbol{x}, \boldsymbol{y}_{<t})}, 1 - \epsilon, 1 + \epsilon \right) A_t \right], \tag{1}$$

where $\boldsymbol{y}$ is sampled from the policy model $\pi_\theta$, $\epsilon$ is a hyperparameter, $\pi_{\theta_{\text{old}}}$ is the old model and the advantage $A_t$ is calculated with generalized advantage estimation (GAE) [62] using rewards and a trained value function $V_\psi$. Although the PPO algorithm aligns human preferences with the policy model outputs effectively, it requires substantial memory and computational resources since the value function is typically another model comparable in size to the policy model.

**Group Relative Policy Optimization (GRPO) [12]** addresses the problem of PPO [60] by approximating the learnable value function with the average reward of multiple sampled outputs. Concretely, given an input sample $\boldsymbol{x}$, the model samples multiple output sequences $\left\{\boldsymbol{y}^{(i)}\right\}_{i=1}^{G}$ from the old policy model $\pi_{\theta_{\text{old}}}$ and then trains the policy model $\pi_\theta$ with the following objective:

$$
\mathcal{L}_{\text{GRPO}}\left(\theta\right) = \mathbb{E}_{\boldsymbol{x}, \left\{\boldsymbol{y}^{(i)}\right\}_{i=1}^{G} \sim \pi_{\theta_{\text{old}}}(\cdot|\boldsymbol{x})}
$$

$$
\frac{1}{|\boldsymbol{y}^{(i)}|} \sum_{t=1}^{|\boldsymbol{y}^{(i)}|} \left\{ \min\left[ \frac{\pi_\theta\left(y_t^{(i)}|\boldsymbol{x},\boldsymbol{y}_{<t}^{(i)}\right)}{\pi_{\theta_{\text{old}}}\left(y_t^{(i)}|\boldsymbol{x},\boldsymbol{y}_{<t}^{(i)}\right)} \hat{A}_t^{(i)}, \text{clip}\left( \frac{\pi_\theta\left(y_t^{(i)}|\boldsymbol{x},\boldsymbol{y}_{<t}^{(i)}\right)}{\pi_{\theta_{\text{old}}}\left(y_t^{(i)}|\boldsymbol{x},\boldsymbol{y}_{<t}^{(i)}\right)}, 1-\epsilon, 1+\epsilon\right) \hat{A}_t^{(i)} \right] \tag{2}
$$

$$
- \beta \mathcal{D}_{\text{KL}}\left[\pi_\theta || \pi_{\text{ref}}\right] \bigg\},
$$

where $\beta$ corresponds to a hyperparameter and $\mathcal{D}_{\text{KL}}$ is the KL-divergence. Here, $\hat{A}^{(i)}$ is advantage calculated based on the relative reward within the group, which is formulated as $\hat{A}^{(i)} = \frac{\mathcal{R}\left(\boldsymbol{x},\boldsymbol{y}^{(i)}\right)-\mu_r}{\sigma_r}$ where $\mu_r, \sigma_r$ denotes the average and standard deviation values of a set of rewards in the group, respectively. Although GRPO has shown its success, GRPO has two limitations that hinder the effective model optimization: *reliance on heuristic constraints* and *vanishing advantage* problems.

**Reliance on safeguards.** GRPO optimizes the model with safeguards implemented using the min and clipping functions to avoid extreme changes in the model. However, the PPO-style clipping function induces **zero gradient** for samples where the value of $\pi_\theta\left(\boldsymbol{y}|\boldsymbol{x}\right)$ is too different from the value of $\pi_{\theta_{\text{old}}}\left(\boldsymbol{y}|\boldsymbol{x}\right)$. It cannot guarantee that the model $\pi_\theta\left(\boldsymbol{y}|\boldsymbol{x}\right)$ stays close to $\pi_{\theta_{\text{ref}}}$ if it is already far from $\pi_{\theta_{\text{ref}}}\left(\boldsymbol{y}|\boldsymbol{x}\right)$ [63]. Similarly, GRPO also suffers from this phenomenon due to the PPO-style hard constraints, and it deteriorates the effective model training. The analysis in [64] also shows that an upper clipping threshold restricts the probability increase of the "exploration" token. This indicates that the safeguards in GRPO negatively influence model optimization.

**Vanishing advantage problem.** The vanishing advantage problem [53] indicates that the advantage of each sample within the group becomes zero, when the rewards of outputs in the group are equal. It is problematic since the model cannot receive any signals from the training sample where the advantage is zero for every response. In particular, we observe that this issue often arises when training samples are either too easy or too difficult for the current model. Training samples with extreme difficulty levels show worse performance than those with moderate difficulty levels.

## 3.2 Regressive GRPO

Here, we present a **Reg-GRPO** (**Reg**ressive **G**roup **R**elative **P**olicy **O**ptimization), which reformulates GRPO into the regression task, removing safeguards such as the min and clipping functions. This reformulation enables the model to directly predict the advantages, resulting in improved alignment of the model with the preference. Following existing RL-based works [65, 66, 42, 45], the Reg-GRPO loss function is derived from the RL objective that maximizes the expected reward with the KL constraints between $\pi_\theta$ and $\pi_{\theta_{\text{old}}}$.

**RL objective.** The objective of our reinforcement learning algorithm for each iteration is to maximize rewards while preventing $\pi_\theta$ from making excessive changes relative to $\pi_{\theta_{\text{old}}}$:

$$
\pi_\theta^* = \arg\max_{\pi_\theta} \mathbb{E}_{\boldsymbol{x},\boldsymbol{y}\sim\pi_\theta(\cdot|\boldsymbol{x})} \mathcal{R}\left(\boldsymbol{x},\boldsymbol{y}\right) - \lambda\, \mathbb{E}_{\boldsymbol{x}}\left[\mathbb{D}_{\text{KL}}\left(\pi_\theta\left(\cdot|\boldsymbol{x}\right)||\pi_{\theta_{\text{old}}}\left(\cdot|\boldsymbol{x}\right)\right)\right], \tag{3}
$$

where $\lambda$ is the hyperparameter that adjusts the strength of the KL-divergence. Following prior works [42, 45], the closed-form solution to the above equation (Eq. (3)) can be obtained by minimum relative entropy problem:

$$
\pi_\theta^*\left(\boldsymbol{y}|\boldsymbol{x}\right) = \frac{1}{Z\left(\boldsymbol{x}\right)} \pi_{\theta_{\text{old}}}\left(\boldsymbol{y}|\boldsymbol{x}\right) \exp\left(\frac{1}{\lambda}\mathcal{R}\left(\boldsymbol{x},\boldsymbol{y}\right)\right), \forall \boldsymbol{x},\boldsymbol{y} \tag{4}
$$

where $Z\left(\boldsymbol{x}\right) = \sum_{\boldsymbol{y}} \pi_{\theta_{\text{old}}}\left(\boldsymbol{y}|\boldsymbol{x}\right) \exp\left(\frac{1}{\lambda}\mathcal{R}\left(\boldsymbol{x},\boldsymbol{y}\right)\right)$ is a partition function. However, since calculating the partition function $Z\left(\boldsymbol{x}\right)$ is expensive, it is hard to obtain $\pi_\theta^*$ exactly.

**Reg-GRPO Loss.** To address these issues, we propose **Reg-GRPO** (**Reg**ressive **GRPO**) loss, which learns the policy model to regress the advantage $\hat{A}^{(i)}$ using the reparameterization, removing the normalization term $Z(\boldsymbol{x})$. Specifically, the advantage for the $i$-th sample is defined as

$$\hat{A}^{(i)} = \frac{\mathcal{R}\left(\boldsymbol{x}, \boldsymbol{y}^{(i)}\right) - \mu_r}{\sigma_r}, \tag{5}$$

where $\mu_r, \sigma_r$ denote the average and standard deviation values of a set of rewards in the group, respectively. We can also rewrite Eq. (4) to express the reward $\mathcal{R}(\boldsymbol{x}, \boldsymbol{y})$ in terms of the optimal model $\pi_\theta^*$, which can be formulated as

$$\mathcal{R}(\boldsymbol{x}, \boldsymbol{y}) = \lambda \cdot \left( \log Z(\boldsymbol{x}) + \log\left( \frac{\pi_\theta^*(\boldsymbol{y}|\boldsymbol{x})}{\pi_{\theta_{\text{old}}}(\boldsymbol{y}|\boldsymbol{x})} \right) \right) \quad \forall \boldsymbol{x}, \boldsymbol{y}. \tag{6}$$

Since the reward can be expressed through the optimal policy $\pi_\theta^*$ (Eq. (6)), the advantage can be equivalently written as $\hat{A}^{(i)} = \frac{\rho^*\left(\boldsymbol{x}, \boldsymbol{y}^{(i)}\right) - \mu_{\rho^*}}{\sigma_{\rho^*}}$, where $\rho^*(\boldsymbol{x}, \boldsymbol{y})$ is defined as $\rho^*(\boldsymbol{x}, \boldsymbol{y}) = \log \frac{\pi_\theta^*(\boldsymbol{y}|\boldsymbol{x})}{\pi_{\theta_{\text{old}}}(\boldsymbol{y}|\boldsymbol{x})}$ and $\mu_{\rho^*}, \sigma_{\rho^*}$ denote mean and standard deviation of $\left\{ \rho^*\left(\boldsymbol{x}, \boldsymbol{y}^{(i)}\right) \right\}_{i=1}^{G}$, respectively. Interestingly, we can see that $Z(\boldsymbol{x})$ is naturally removed during the reformulation.

Building on this insight, we define the predictive advantage, which estimates the advantage calculated by normalizing rewards within a group of samples, using the current policy $\pi_\theta$ as

$$\hat{A}_\theta^{(i)} = \frac{\rho\left(\boldsymbol{x}, \boldsymbol{y}^{(i)}\right) - \mu_\rho}{\sigma_\rho}, \quad \rho(\boldsymbol{x}, \boldsymbol{y}) = \log \frac{\pi_\theta(\boldsymbol{y}|\boldsymbol{x})}{\pi_{\theta_{\text{old}}}(\boldsymbol{y}|\boldsymbol{x})}, \tag{7}$$

where $\mu_\rho, \sigma_\rho$ are mean and standard deviation of $\left\{ \rho\left(\boldsymbol{x}, \boldsymbol{y}^{(i)}\right) \right\}_{i=1}^{G}$, respectively. Then, we optimize the policy by minimizing the gap between the target advantage $\hat{A}$ and its predicted counterpart $\hat{A}_\theta$ using Reg-GRPO (Regressive GRPO), which is defined as:

$$\mathcal{L}_{\text{Reg-GRPO}}(\theta) = \mathbb{E}_{\boldsymbol{x}, \left\{\boldsymbol{y}^{(i)}\right\}_{i=1}^{G} \sim \pi_{\theta_{\text{old}}}(\cdot|\boldsymbol{x})} \left\{ \left( \hat{A}^{(i)} - \hat{A}_\theta^{(i)} \right)^2 - \beta\, \mathbb{D}_{\text{KL}}\left[\pi_\theta || \pi_{\text{ref}}\right] \right\}, \tag{8}$$

Similar to GRPO, we regularize the update with the KL divergence. The proposed Reg-GRPO loss serves as an effective alternative for optimizing group-level objectives, showing better performance than GRPO. In our experiments, we demonstrate that this formulation leads to faster convergence and improved policy quality. The detailed derivation procedure of Reg-GRPO is in Appendix A.

### 3.3 Difficulty-aware data augmentation

In this section, we present a **difficulty-aware data augmentation** framework, which addresses the vanishing advantage problem in GRPO. This issue arises when training samples are either too easy or too difficult, leading to uniform rewards across multiple responses. As a result, the advantage values become zero, erasing the learning signal. Our augmentation strategy mitigates this issue by modulating the difficulty of inputs to increase variance in predicted rewards, thereby preserving informative gradients for effective model optimization.

Specifically, given an input sample $\boldsymbol{x} = (\boldsymbol{v}, \boldsymbol{q})$, where $\boldsymbol{v}$ and $\boldsymbol{q}$ denote a video and a question, respectively, we first generate multiple responses and then compute the average reward $\frac{1}{G} \sum_{i=1}^{G} \mathcal{R}\left(\boldsymbol{x}, \boldsymbol{y}^{(i)}\right)$ for the sample. We then measure the difficulty of $\boldsymbol{x}$ by comparing its average reward with the average reward of samples in a replay buffer $\mathbb{B}^W$, which consists of samples $\boldsymbol{x}_{\text{rep}}$ and their corresponding outputs $\left\{ \boldsymbol{y}_{\text{rep}}^{(i)} \right\}_{i=1}^{G}$ from the most recent $W$ steps. Formally, the difficulty $\Delta_\mathcal{R}(\boldsymbol{x})$ of the input sample $\boldsymbol{x}$ is calculated as:

$$\Delta_\mathcal{R}(\boldsymbol{x}) = \mathbb{E}_{\left(\boldsymbol{x}_{\text{rep}}, \left\{\boldsymbol{y}_{\text{rep}}^{(i)}\right\}_{i=1}^{G}\right) \in \mathbb{B}^W} \left[ \frac{1}{G} \sum_{i=1}^{G} \mathcal{R}\left(\boldsymbol{x}_{\text{rep}}, \boldsymbol{y}_{\text{rep}}^{(i)}\right) \right] - \frac{1}{G} \sum_{j=1}^{G} \mathcal{R}\left(\boldsymbol{x}, \boldsymbol{y}^{(j)}\right). \tag{9}$$

Instead of using only the sample reward $\mathcal{R}(\boldsymbol{x}, \boldsymbol{y})$, we use the average reward of samples in the replay buffer as a reference value to account for model evolution.

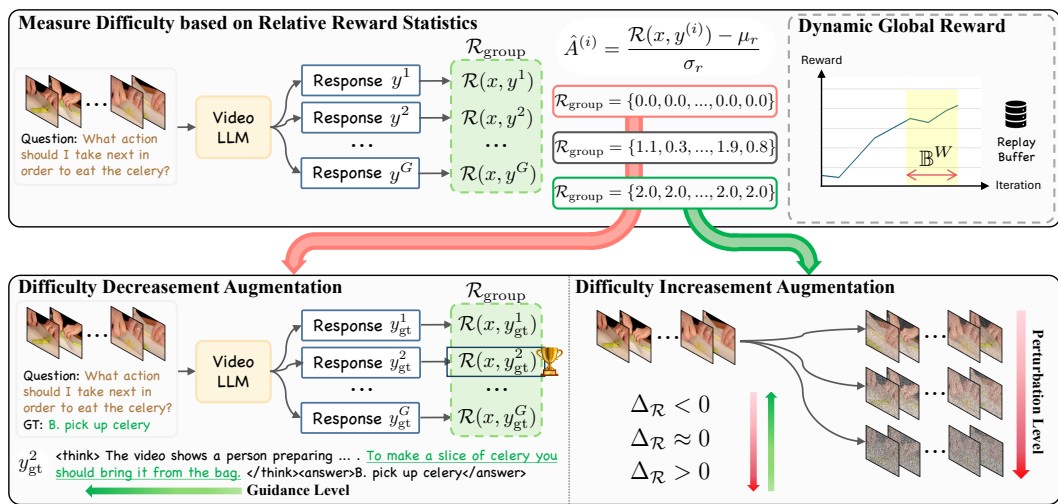

Figure 2: **Overview of the difficulty-aware data augmentation.** First, we assess the difficulty of responses given the input video and question using Eq. (9). For hard samples, it augments the input prompts with the reasoning cues extracted from successful reasoning paths (Difficulty-decreasing augmentation), while the easy samples are perturbed with the noise (Difficulty-increasing augmentation). The scale of the guidance level or noise level is adaptively determined based on the difficulty of the current sample.

Building on the difficulty metric in Eq. (9), we adaptively adjust each training sample to balance the learning signal. For samples identified as too easy (*i.e.*, high reward and low difficulty), we increase task complexity by perturbing the input—for example, by adding Gaussian noise to video inputs or masking video frames—to encourage the model to attend to more informative cues. Conversely, for too difficult samples (*i.e.*, low reward and high difficulty), we inject auxiliary reasoning hints or visual cues that simplify the temporal context, helping the model focus on core reasoning paths instead of failing due to overwhelming difficulty. This dynamic modulation leads to an appropriate difficulty distribution across training, preventing the collapse of advantage values while ensuring that each update provides a meaningful gradient signal.

**Difficulty-decreasing augmentation.** For difficult samples ($\Delta_{\mathcal{R}}(\boldsymbol{x}) > 0$), we ease the difficulty of the sample by providing auxiliary reasoning cues that guide the model toward generating correct reasoning. Concretely, we first augment the input prompt $\boldsymbol{q}$ with the ground-truth answer and generate multiple reasoning trajectories using the VideoLLM. Among these reasoning trajectories, we select the response $\boldsymbol{y}_{\mathrm{gt}}$ with the highest reward and extract a partial reasoning trace of it. This trace is then incorporated into the original prompt to form a modified prompt $\tilde{\boldsymbol{q}}$ containing structured hints that guide the model's reasoning toward the correct solution. To maintain adaptive control, the guidance level is scaled according to the sample's difficulty magnitude. Harder samples receive stronger reasoning cues, while moderately difficult samples are given lighter guidance. By adaptively providing guidance for challenging inputs, this augmentation mitigates vanishing gradients in difficult cases and facilitates more stable convergence through progressively refined reasoning.

**Difficulty-increasing augmentation.** Conversely, for easy samples ($\Delta_{\mathcal{R}}(\boldsymbol{x}) < 0$), we employ a difficulty-increasing augmentation to enhance task complexity and encourage the model to explore more diverse reasoning trajectories. We perturb the visual input $\boldsymbol{v}$ to create a harder input $\tilde{\boldsymbol{v}}$ by introducing frame-level Gaussian noise, thereby slightly degrading perceptual fidelity while preserving overall semantic structure. The intensity of the noise is proportionally scaled by the difficulty magnitude, ensuring that easier samples receive stronger perturbations and moderately easy samples remain stable. This adaptive corruption expands the diversity of generated reasoning trajectories. By inducing distributed rewards, the augmentation ensures that even trivial samples provide informative gradients, avoiding zero learning signals throughout the optimization process.

Table 1: **Performance on SEED-Bench-R1 validation split and LongVideoBench.** In-Dist. means in-distribution dataset.

| Method | SBR-L1 In-Dist. | SBR-L2 Cross-Env | SBR-L3 Cross-Task, Cross-Env | | | | | LongVideoBench Cross-Task, Cross-Env | | | | |
|---|---|---|---|---|---|---|---|---|---|---|---|---|
| | Daily life | Daily life | Daily life | Hobbies | Recreation | Work | Overall | (8,15] | (15,60] | (180,600] | (900,3600] | Overall |
| *Baseline* | | | | | | | | | | | | |
| VideoLLaMA3-7B [9] | 33.3 | 33.2 | 26.7 | 28.5 | 30.6 | 27.0 | 27.7 | 35.7 | 43.1 | 21.0 | 22.5 | 26.7 |
| InternVL3-2B [8] | 23.7 | 23.1 | 21.2 | 16.3 | 18.6 | 12.6 | 17.1 | 41.6 | 48.4 | 33.7 | 30.0 | 34.8 |
| InternVL3-8B [8] | 41.4 | 40.8 | 39.2 | 34.6 | 35.0 | 30.0 | 34.8 | 54.6 | 66.7 | 46.1 | 44.2 | 49.1 |
| InternVL3-14B [8] | 43.6 | 45.8 | 44.0 | 34.9 | 36.6 | 31.7 | 37.2 | 69.2 | 62.8 | 46.3 | 42.0 | 50.0 |
| *Qwen2-VL-2B* | | | | | | | | | | | | |
| Qwen2-VL-2B [5] | 12.9 | 16.4 | 13.0 | 16.3 | 10.4 | 14.4 | 13.8 | 33.0 | 32.7 | 29.5 | 22.2 | 27.2 |
| SFT | 34.1 | 36.2 | 36.9 | 34.6 | 30.0 | 33.9 | 33.8 | 42.6 | 42.3 | 37.0 | 32.2 | 36.3 |
| GRPO | 38.4 | 42.0 | 40.6 | 37.6 | 30.5 | 40.4 | 36.8 | 47.0 | 46.4 | 36.6 | 32.1 | 37.4 |
| **DeepVideo-R1** | **48.9** | **50.3** | **52.4** | **42.7** | **41.1** | **49.2** | **46.3** | **51.4** | **49.7** | **38.5** | **34.8** | **40.1** |
| *Qwen2-VL-7B* | | | | | | | | | | | | |
| Qwen2-VL-7B [5] | 34.8 | 34.0 | 31.2 | 32.3 | 33.3 | 30.7 | 31.6 | 42.3 | 44.8 | 34.0 | 25.4 | 32.9 |
| SFT | 43.8 | 44.1 | 38.3 | 41.0 | 32.2 | 38.6 | 38.2 | 45.0 | 54.7 | 36.7 | 36.4 | 40.0 |
| GRPO | 46.0 | 50.2 | 48.5 | 45.1 | 43.7 | 41.3 | 44.9 | 54.5 | 53.5 | 42.5 | 37.2 | 43.4 |
| **DeepVideo-R1** | **56.4** | **59.8** | **57.6** | **52.5** | **50.0** | **55.2** | **53.8** | **56.2** | **61.4** | **45.1** | **40.8** | **46.9** |
| *Qwen2.5-VL-3B* | | | | | | | | | | | | |
| Qwen2.5-VL-3B [7] | 31.3 | 32.7 | 33.0 | 28.8 | 27.3 | 23.0 | 28.2 | 50.3 | 62.1 | 39.0 | 32.1 | 40.4 |
| SFT | 35.9 | 39.1 | 39.9 | 31.5 | 29.7 | 31.2 | 33.7 | 51.4 | 52.9 | 36.3 | 35.3 | 39.7 |
| GRPO | 39.6 | 41.0 | 39.0 | 33.9 | 36.6 | 31.9 | 35.4 | 51.4 | 62.1 | 42.2 | 36.0 | 43.2 |
| **DeepVideo-R1** | **48.1** | **51.1** | **46.5** | **45.8** | **43.7** | **40.1** | **44.0** | **54.1** | **64.1** | **45.9** | **43.4** | **48.4** |
| *Qwen2.5-VL-7B* | | | | | | | | | | | | |
| Qwen2.5-VL-7B [7] | 33.4 | 38.2 | 35.1 | 31.5 | 27.3 | 28.0 | 31.0 | 54.6 | **63.4** | 37.8 | 36.2 | 42.5 |
| SFT | 42.4 | 42.6 | 41.2 | 37.3 | 36.6 | 41.5 | 39.0 | 54.6 | 56.2 | 41.5 | 38.1 | 43.9 |
| GRPO | 49.1 | 52.1 | 49.4 | 40.7 | 43.2 | 35.2 | 42.2 | 61.1 | 60.8 | 44.4 | 40.8 | 47.7 |
| **DeepVideo-R1** | **52.0** | **55.7** | **51.3** | **47.8** | **47.0** | **44.1** | **47.7** | **62.7** | **63.4** | **49.3** | **44.5** | **51.1** |

# 4 Experiments

## 4.1 Experimental Settings

To validate the effectiveness of the proposed method, we conduct evaluations on various video benchmarks, including both general video understanding tasks (*e.g.*, SEED-Bench-R1 [56], VSI-Bench, Video-MMMU, MMVU (mc), MVBench, TempCompass, Video-MME (wo sub)), long video understanding tasks (*e.g.*, LongVideoBench [25]), and fine-grained spatial-temporal video reasoning tasks (NExTGQA [37]). More details about datasets are in Appendix E. We employ Qwen2-VL-2B/7B [5] and Qwen2.5-VL-3B/7B [7] for the experiments. For the analysis, we use Qwen2.5-VL-3B as a default video LLM. More implementation details are in Appendix C.1.

## 4.2 Experimental Results

**Experimental results on SEED-Bench-R1.** Table 1 summarizes the performance of various baselines, supervised fine-tuning (SFT), GRPO, and our proposed DeepVideo-R1 on the validation splits of the SEED-Bench-R1 (SBR) dataset. Across all settings (SBR-L1, L2, L3), DeepVideo-R1 consistently achieves the best performance, demonstrating its strong capability for video reasoning under both in-distribution and cross-environment settings. Specifically, compared with Qwen2.5-VL-3B + GRPO, our DeepVideo-R1-3B improves the overall scores on SBR-L1, L2, and L3 by +8.5, +10.1, and +8.6 points, respectively. Notably, the performance gains on SBR-L2 and L3 (overall) exceed those on SBR-L1, indicating that DeepVideo-R1 enhances generalization across cross-task and cross-environment settings. These results suggest that the proposed regression-based optimization and reasoning-aware augmentation in DeepVideo-R1 enable more stable policy learning and improved adaptability to diverse video understanding scenarios.

**Experimental results on LongVideoBench.** We further evaluate DeepVideo-R1 on LongVideoBench, a benchmark designed to assess long-video reasoning and temporal compositional understanding. As shown in Table 1, DeepVideo-R1 again outperforms all baselines across varying temporal ranges, achieving an overall score of 51.1, surpassing both SFT and GRPO-trained counterparts. In particular, DeepVideo-R1-3B achieves a substantial improvement of +7.4 over Qwen2.5-VL-3B + GRPO on the longest duration range (900 – 3600 s), underscoring its superior ability to reason over extended temporal contexts. This strong performance on long-duration videos highlights DeepVideo-R1's effectiveness in maintaining coherent reasoning over time, validating its robustness in complex real-world video understanding tasks.

Table 2: **Performance on various video reasoning and general benchmarks.**

| Method | Video Reasoning Benchmark | | | Video General Benchmark | | |
|---|---|---|---|---|---|---|
| | VSI-Bench | Video-MMMU | MMVU (mc) | MVBench | TempCompass | Video-MME (wo sub) |
| LLaMA-VID [67] | - | - | - | 41.9 | 45.6 | - |
| VideoLLaMA2 [33] | - | - | 44.8 | 54.6 | - | 47.9 |
| LongVA-7B [68] | 29.2 | 23.9 | - | - | 56.9 | 52.6 |
| VILA-1.5-8B [69] | 28.9 | 20.8 | - | - | 58.8 | - |
| VILA-1.5-40B [69] | 31.2 | 34.0 | - | - | - | 60.1 |
| Video-UTR-7B [70] | - | - | - | 58.8 | 59.7 | 52.6 |
| LLaVA-OneVision-7B [31] | 32.4 | 33.8 | 49.2 | 56.7 | - | 58.2 |
| Kangaroo-8B [71] | - | - | - | 61.1 | 62.5 | 56.0 |
| Qwen2.5-VL-3B [7] | 32.4 | 36.1 | 54.2 | 48.1 | 29.7 | **54.4** |
| **DeepVideo-R1-3B (Ours)** | **33.0** | **40.7** | **59.0** | **49.6** | **63.1** | 51.1 |

Table 3: **Experimental results on NExTGQA**

| Method | mIoU | Acc@QA |
|---|---|---|
| *Vision Experts* | | |
| IGV [72] | 14.0 | 50.1 |
| Temp[CLIP] [37] | 12.1 | 60.2 |
| FrozenBiLM [73] | 9.6 | 70.8 |
| SeViLA [74] | 21.7 | 68.1 |
| *VideoLLMs* | | |
| VideoChat-R1 [55] | 32.4 | 70.6 |
| VideoChat-R1-thinking [55] | 36.1 | 69.2 |
| **DeepVideo-R1-7B (Ours)** | **36.8** | **72.5** |

Table 4: **Ablation study** on training schemes (Reg-GRPO and difficulty-aware data augmentation (DA-Aug.) in DeepVideo-R1 using SEED-Bench-R1 dataset.

| Method | DA-Aug. | L1 (In-Dist.) | L2 (OOD) | L3 (OOD) |
|---|---|---|---|---|
| Qwen2.5-VL-3B | | 31.3 | 32.7 | 27.0 |
| GRPO | | 39.6 | 41.0 | 35.4 |
| GRPO | ✓ | 41.7 | 42.5 | 36.6 |
| Reg-GRPO | | 44.2 | 44.2 | 39.5 |
| Reg-GRPO | ✓ | **48.1** | **51.1** | **44.0** |

**Experimental results on various video benchmarks.** We further evaluate DeepVideo-R1-3B, built upon Qwen2.5-VL-3B, on diverse video reasoning and general benchmarks to assess its generalization ability. Following [54], we use the Video-R1 [54] training set to train the model, and report the results in Table 2. As shown in the table, DeepVideo-R1-3B consistently outperforms Qwen2.5-VL-3B and other large-scale multimodal video models (e.g., LLAVA-OneVision-7B, VILA-1.5-40B, Kangaroo-8B) across almost all benchmarks. In comparison of the experimental results of the base model Qwen2.5-VL-7B, our DeepVideo-R1 consistently improves the performance on 5 out of 6 datasets. In particular, DeepVideo-R1 improves performance from 29.7 to 63.1 on TempCompass. Overall, these results confirm that DeepVideo-R1 effectively generalizes beyond SEED-Bench-R1, establishing a new performance level across both reasoning-oriented and general video understanding benchmarks.

**Experimental results on NExTGQA.** Table 3 reports the performance of DeepVideo-R1 compared with both vision experts (IGV, Temp[CLIP], FrozenBiLM, SeViLA) and VideoLLMs (VideoChat-R1, VideoChat-R1-thinking). For the NExT-GQA benchmark, DeepVideo-R1-7B is trained with a composite reward combining accuracy, format consistency, and IoU, aligning with the dataset's grounding-oriented evaluation. As shown, DeepVideo-R1-7B achieves 36.8 mIoU and 72.5 Acc@QA, outperforming all baselines, including a +4.4 mIoU and +2.3 Acc@QA gain over VideoChat-R1. These improvements highlight DeepVideo-R1's ability to enhance both spatial grounding and reasoning precision, demonstrating its robustness to diverse reward designs and its strong adaptability across grounded video reasoning tasks.

### 4.3 Analysis

**Ablation studies.** We conduct ablation studies to explore the contribution of Reg-GRPO and difficulty-aware data augmentation (DA-Aug.) in Table 4. The table demonstrates that both Reg-GRPO and difficulty-aware data augmentation contribute to the performance improvement of DeepVideo-R1. By comparing the GRPO-trained model without and with difficulty-aware data augmentation, we observe a +2.1-point improvement on SBR (L1), suggesting that difficulty-aware data augmentation is effective for both GRPO and Reg-GRPO. Also, Reg-GRPO (w/o DA-Aug.) yields a +4.1-point improvement over Qwen2.5-VL-3B+GRPO (w/o DA-Aug.) on SBR (L3). This indicates that Reg-GRPO is more effective than GRPO by directly predicting advantages.

Table 5: Performance comparison on reinforcement learning algorithm.

| Method | L1 | L2 | L3 |
|---|---|---|---|
| Qwen2.5-VL-3B | 31.3 | 32.7 | 27.0 |
| + DPO [42] | 35.8 | 35.2 | 30.8 |
| + Online DPO [42] | 37.1 | 38.1 | 31.9 |
| + REINFORCE [75–77] | 37.0 | 39.5 | 32.3 |
| + RLOO [78] | 35.0 | 37.4 | 31.3 |
| + REBEL [45] | 41.8 | 43.7 | 38.0 |
| + Reward-Regression | 32.5 | 33.1 | 28.3 |
| + GRPO [12, 13] | 39.6 | 41.0 | 35.4 |
| **+ Reg-GRPO (Ours)** | **44.2** | **44.2** | **39.5** |

Table 6: Performance comparison on absolute and relative difficulty measurement.

| Diff. ref. | L1 | L2 | L3 |
|---|---|---|---|
| Absolute | 47.9 | 50.1 | 40.7 |
| Relative | **48.1** | **51.1** | **44.0** |

Table 7: Performance comparison according to the data augmentation types. Diff↑ indicates difficulty-increasing augmentation. Diff↓ indicates difficulty-decreasing augmentation.

| Diff. ↑ | Diff. ↓ | L1 | L2 | L3 |
|---|---|---|---|---|
| | | 44.2 | 44.2 | 36.3 |
| ✓ | | 45.3 | 46.9 | 40.0 |
| | ✓ | 45.3 | 47.3 | 41.6 |
| ✓ | ✓ | **48.1** | **51.1** | **44.0** |

Table 8: Performance comparison on augmentation scaling scheme (Fixed guidance/noise level v.s. Adaptive guidance/noise level).

| Aug. | L1 | L2 | L3 |
|---|---|---|---|
| No Aug. | 44.2 | 44.2 | 36.3 |
| Fixed | 46.8 | 48.6 | 43.0 |
| Adaptive | **48.1** | **51.1** | **44.0** |

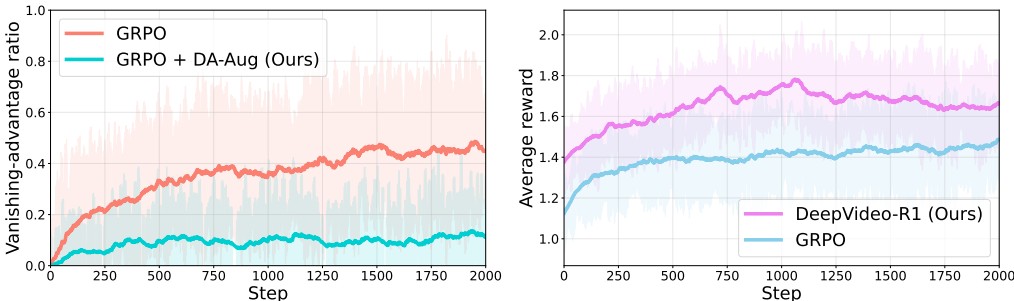

Figure 3: **Vanishing advantage ratio comparison** on GRPO and GRPO+DA-Aug (Difficulty-aware augmentation) (Left). **Reward curves** of DeepVideo-R1 (Ours) and GRPO (Right).

**Comparison with reinforcement learning methods.** Table 5 compares our Reg-GRPO with representative reinforcement fine-tuning (RFT) methods. The detailed explanations about the reinforcement fine-tuning methods are in Appendix D. From the table, our DeepVideo-R1 achieves the best performance among the compared RFT methods. In particular, compared to reward regression, which directly regresses reward scores, DeepVideo-R1 performs significantly better. This result highlights that directly aligning advantages is more effective than reward regression.

**Impact of relative difficulty measurement.** We compare absolute and relative difficulty measurements for adaptive data augmentation in Table 6. The relative scheme, which considers reward history statistics, consistently outperforms the absolute counterpart across all difficulty levels (L1–L3) on the SEED-Bench-R1 dataset, demonstrating its superior ability to guide effective augmentation.

**Impact of difficulty-decreasing/increasing augmentation.** In Table 7, we compare different data augmentation schemes: difficulty-increasing and difficulty-decreasing augmentations. The table demonstrates that the model trained with both schemes achieves the best performance, yielding a +7.7-point gain on SBR (L3), which is an out-of-distribution dataset. This reveals that adjusting the sample's difficulty to a moderate level is important for learning with group-normalized advantages.

**Impact of augmentation scaling scheme.** In Table 8, we compare different augmentation scaling strategies: no augmentation, fixed scaling (constant guidance/noise level), adaptive scaling (difficulty-aware guidance/noise). From the table, the adaptive strategy consistently outperforms the others across all difficulty levels (L1–L3), demonstrating the importance of tailoring augmentation strength based on input difficulty.

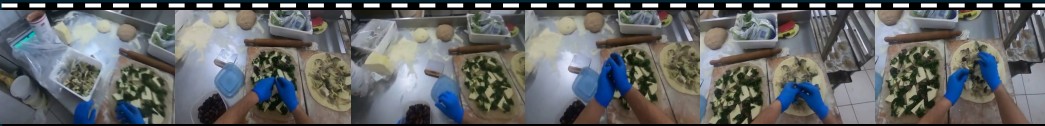

**Question:** Considering the progress shown in the video and my current observation shown in the image, what action should I take next in order to prepare berries?

A. move tray     B. close tray     C. open container     **D. move berries**

Output the thinking process in <think> </think> and final answer in <answer> </answer> tags, i.e., <think> reasoning process here </think><answer> answer here </answer>.

**GRPO:** <think> The image shows a person preparing food, specifically making a dish with vegetables and cheese on a flatbread. **There are no berries or trays visible in the image**, so it is not possible to determine the next steps for preparing berries. The options provided are related to trays and containers, which are not relevant to the current task of preparing berries. </think> <answer> **C. open container** </answer>

**DeepVideo-R1(Ours):** <think> The person in the video is currently preparing a dish with berries. The next logical step would be **to add more berries to the dish or to arrange the existing berries in a specific way**. Since the image shows the person already handling the berries, the next action should be to **add more berries or to arrange the existing ones.** </think> <answer> **D. move berries** </answer>

Figure 4: **Qualitative result of DeepVideo-R1-7B** in comparison of Qwen2.5-VL-7B+GRPO.

**Impact of augmentation.**    We compare the vanishing advantage ratio of GRPO and GRPO with DA-Aug (difficulty-aware data augmentation) in Figure 3 (left). The figure shows that our data augmentation effectively reduces the ratio of samples causing the vanishing advantage. This indicates that the data augmentation effectively addresses the vanishing advantage problem in GRPO.

**Reward curves.**    In addition, we plot the reward curves of GRPO and our DeepVideo-R1, where the x-axis is training step and y-axis is the average reward in Figure 3 (right). From the figure, our DeepVideo-R1 gets a higher average reward with Reg-GRPO and difficulty-aware data augmentation.

**Qualitative results.**    Figure 4 presents a qualitative example from SEED-Bench-R1-7B, comparing the outputs of our DeepVideo-R1 and the Qwen2.5-VL-7B trained with GRPO. The task is to predict the next action given a video. Our DeepVideo-R1 correctly infers that the person will continue moving berries. While the GRPO-only model fails to recognize the presence of berries. This demonstrates that DeepVideo-R1 has strong visual grounding and understanding capabilities.

## 5    Conclusion

We propose a video large language model, DeepVideo-R1, trained with Reg-GRPO (Regressive GRPO) and difficulty-aware data augmentation to address two key challenges in group relative policy optimization. Reg-GRPO reformulates the GRPO objective as a regression task that directly aligns the model with group-normalized advantages. Difficulty-aware data augmentation modulates input difficulty to alleviate the vanishing advantage problem. Our experiments demonstrate that DeepVideo-R1 consistently improves the reasoning performance of diverse VideoLLMs, outperforming GRPO-based reinforcement fine-tuning.

## Acknowledgment

This work partly supported by Korea Research Institute for defense Technology planning and advancement - Grant funded by Defense Acquisition Program Administration(DAPA)(KRIT-CT-23-021, 30%), the InnoCORE program of the Ministry of Science and ICT(N10250156, 30%), Virtual Engineering Platform Project (Grant No. P0022336, 30%), funded by the Ministry of Trade, Industry & Energy (MoTIE, South Korea), and Electronics and Telecommunications Research Institute(ETRI) grant funded by the Korean government [25ZB1200, Fundamental Technology Research for Human-Centric Autonomous Intelligent Systems, 10%].

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

# A Discussion and Derivation of Reg-GRPO

## A.1 Deriving the optimum of the KL-constrained reward maximization optimization

We derive Eq. (4) from Eq. (3) following [42]. We optimize $\pi_\theta$ with the following objective:

$$\pi_\theta^* = \arg\max_{\pi_\theta} \mathbb{E}_{\boldsymbol{x}, \boldsymbol{y} \sim \pi_\theta(\cdot|\boldsymbol{x})} \mathcal{R}(\boldsymbol{x}, \boldsymbol{y}) - \lambda \, \mathbb{E}_{\boldsymbol{x}} \left[ \mathbb{D}_{\text{KL}} \left( \pi_\theta(\cdot|\boldsymbol{x}) \, || \, \pi_{\theta_{\text{old}}}(\cdot|\boldsymbol{x}) \right) \right], \tag{10}$$

where $\mathcal{R}$ is the reward function, $\pi_{\theta_{\text{old}}}$ is the old policy model, and $\lambda$ ($\lambda \geq 0$) denotes a hyperparameter. We can obtain a closed-form solution to the above relative-entropy-regularized maximization problem.

$$
\begin{aligned}
\pi_\theta^* &= \arg\max_{\pi_\theta} \; \mathbb{E}_{\boldsymbol{x}, \boldsymbol{y} \sim \pi_\theta(\cdot|\boldsymbol{x})} \mathcal{R}(\boldsymbol{x}, \boldsymbol{y}) - \lambda \, \mathbb{E}_{\boldsymbol{x}} \left[ \mathbb{D}_{\text{KL}} \left( \pi_\theta(\cdot|\boldsymbol{x}) \, || \, \pi_{\theta_{\text{old}}}(\cdot|\boldsymbol{x}) \right) \right] \\
&= \arg\max_{\pi_\theta} \mathbb{E}_{\boldsymbol{x}, \boldsymbol{y} \sim \pi_\theta(\cdot|\boldsymbol{x})} \left[ \mathcal{R}(\boldsymbol{x}, \boldsymbol{y}) - \lambda \cdot \log \frac{\pi_\theta(\boldsymbol{y}|\boldsymbol{x})}{\pi_{\theta_{\text{old}}}(\boldsymbol{y}|\boldsymbol{x})} \right] \\
&= \arg\min_{\pi_\theta} \mathbb{E}_{\boldsymbol{x}, \boldsymbol{y} \sim \pi_\theta(\cdot|\boldsymbol{x})} \left[ \log \frac{\pi_\theta(\boldsymbol{y}|\boldsymbol{x})}{\pi_{\theta_{\text{old}}}(\boldsymbol{y}|\boldsymbol{x})} - \frac{1}{\lambda} \cdot \mathcal{R}(\boldsymbol{x}, \boldsymbol{y}) \right] \\
&= \arg\min_{\pi_\theta} \mathbb{E}_{\boldsymbol{x}, \boldsymbol{y} \sim \pi_\theta(\cdot|\boldsymbol{x})} \left[ \log \frac{\pi_\theta(\boldsymbol{y}|\boldsymbol{x})}{\pi_{\theta_{\text{old}}}(\boldsymbol{y}|\boldsymbol{x}) \exp\left(\frac{1}{\lambda}\mathcal{R}(\boldsymbol{x}, \boldsymbol{y})\right)} \right] \\
&= \arg\min_{\pi_\theta} \mathbb{E}_{\boldsymbol{x}, \boldsymbol{y} \sim \pi_\theta(\cdot|\boldsymbol{x})} \left[ \log \frac{\pi_\theta(\boldsymbol{y}|\boldsymbol{x})}{\frac{1}{Z(\boldsymbol{x})} \pi_{\theta_{\text{old}}}(\boldsymbol{y}|\boldsymbol{x}) \exp\left(\frac{1}{\lambda}\mathcal{R}(\boldsymbol{x}, \boldsymbol{y})\right)} - \log Z(\boldsymbol{x}) \right],
\end{aligned}
\tag{11}
$$

where $Z(\boldsymbol{x}) = \sum_{\boldsymbol{y}} \pi_{\theta_{\text{old}}}(\boldsymbol{y}|\boldsymbol{x}) \exp\left(\frac{1}{\lambda}\mathcal{R}(\boldsymbol{x}, \boldsymbol{y})\right)$ is a partition function. Please note that the partition function is only dependent on $\boldsymbol{x}$ and the old policy $\pi_{\theta_{\text{old}}}$.

Now let $\bar{\pi}_\theta$ be defined as:

$$\bar{\pi}_\theta(\boldsymbol{y}|\boldsymbol{x}) = \frac{1}{Z(\boldsymbol{x})} \pi_{\theta_{\text{old}}}(\boldsymbol{y}|\boldsymbol{x}) \exp\left(\frac{1}{\lambda}\mathcal{R}(\boldsymbol{x}, \boldsymbol{y})\right). \tag{12}$$

It can be seen as a valid probability distribution as $\bar{\pi}_\theta(\boldsymbol{y}|\boldsymbol{x}) \geq 0$ for all $\boldsymbol{y}$ and $\sum_{\boldsymbol{y}} \bar{\pi}_\theta(\boldsymbol{y}|\boldsymbol{x}) = 1$. Since $Z(\boldsymbol{x})$ is not a function of $\boldsymbol{y}$, the above minimization problem can be formulated as

$$
\begin{aligned}
&\arg\min_{\pi} \mathbb{E}_{\boldsymbol{x}, \boldsymbol{y} \sim \pi_\theta(\cdot|\boldsymbol{x})} \left[ \log \frac{\pi_\theta(\boldsymbol{y}|\boldsymbol{x})}{\bar{\pi}_\theta(\boldsymbol{y}|\boldsymbol{x})} - \log Z(\boldsymbol{x}) \right] \\
&= \arg\min_{\pi} \mathbb{E}_{\boldsymbol{x}} \left[ \mathbb{D}_{\text{KL}} \left( \pi_\theta(\boldsymbol{y}|\boldsymbol{x}) \, || \, \bar{\pi}_\theta(\boldsymbol{y}|\boldsymbol{x}) \right) - \log Z(\boldsymbol{x}) \right]
\end{aligned}
\tag{13}
$$

Since the partition function $Z(\boldsymbol{x})$ is not dependent on $\pi$, the optimal $\pi^*$ is the policy that minimizes the first KL term. Since the optimal KL-divergence is achieved if and only if two distributions are identical, we have optimal solution as:

$$\pi_\theta^*(\boldsymbol{y}|\boldsymbol{x}) = \frac{1}{Z(\boldsymbol{x})} \pi_{\theta_{\text{old}}}(\boldsymbol{y}|\boldsymbol{x}) \exp\left(\frac{1}{\lambda}\mathcal{R}(\boldsymbol{x}, \boldsymbol{y})\right), \quad \forall \boldsymbol{x}, \boldsymbol{y}. \tag{14}$$

## A.2 Deriving the reward in terms of the optimal policy

The reward can be reorganized under the optimal policy. We can invert Eq. (14) as follows:

$$
\begin{aligned}
\exp\left(\frac{1}{\lambda}\mathcal{R}(\boldsymbol{x}, \boldsymbol{y})\right) &= Z(\boldsymbol{x}) \frac{\pi_\theta^*(\boldsymbol{y}|\boldsymbol{x})}{\pi_{\theta_{\text{old}}}(\boldsymbol{y}|\boldsymbol{x})}, \\
\frac{1}{\lambda}\mathcal{R}(\boldsymbol{x}, \boldsymbol{y}) &= \log Z(\boldsymbol{x}) + \log\left(\frac{\pi_\theta^*(\boldsymbol{y}|\boldsymbol{x})}{\pi_{\theta_{\text{old}}}(\boldsymbol{y}|\boldsymbol{x})}\right), \\
\mathcal{R}(\boldsymbol{x}, \boldsymbol{y}) &= \lambda \cdot \left(\log Z(\boldsymbol{x}) + \log\left(\frac{\pi_\theta^*(\boldsymbol{y}|\boldsymbol{x})}{\pi_{\theta_{\text{old}}}(\boldsymbol{y}|\boldsymbol{x})}\right)\right), \quad \forall \boldsymbol{x}, \boldsymbol{y}.
\end{aligned}
\tag{15}
$$

## A.3 Deriving the advantage in terms of the optimal policy.

The advantage $\hat{A}^{(i)}$ is defined as

$$\hat{A}^{(i)} = \frac{\mathcal{R}\left(\boldsymbol{x}, \boldsymbol{y}^{(i)}\right) - \mu_r}{\sigma_r}, \tag{16}$$

where $\mu_r, \sigma_r$ denotes the average and standard deviation values of a set of rewards in the group, respectively. We can rewrite the advantage in terms of the optimal policy in Eq. (15) as follows:

$$\begin{aligned}
\hat{A}^{(i)} &= \frac{\rho^*\left(\boldsymbol{x}, \boldsymbol{y}^{(i)}\right) + \log Z\left(\boldsymbol{x}\right) - \left(\frac{1}{G}\sum_{j=1}^{G}\rho^*\left(\boldsymbol{x}, \boldsymbol{y}^{(j)}\right) + \log Z\left(\boldsymbol{x}\right)\right)}{\sigma_{\rho^*}}, \\
&= \frac{\rho^*\left(\boldsymbol{x}, \boldsymbol{y}^{(i)}\right) - \mu_{\rho^*}}{\sigma_{\rho^*}}, \quad \rho^*\left(\boldsymbol{x}, \boldsymbol{y}\right) = \log\frac{\pi_\theta^*\left(\boldsymbol{y}|\boldsymbol{x}\right)}{\pi_{\theta_{\text{old}}}\left(\boldsymbol{y}|\boldsymbol{x}\right)},
\end{aligned} \tag{17}$$

where $\mu_{\rho^*}, \sigma_{\rho^*}$ are mean and standard deviation of $\left\{\rho^*\left(\boldsymbol{x}, \boldsymbol{y}^{(i)}\right)\right\}_{i=1}^{G}$, respectively. Interestingly, we can see that $Z\left(\boldsymbol{x}\right)$ is removed.

## A.4 Reg-GRPO

Based on Eq. (17), our Reg-GRPO (Regressive GRPO) is to learn the model $\pi_\theta$ to directly predict the advantage as follows:

$$\begin{aligned}
\mathcal{L}_{\text{Reg-GRPO}}\left(\theta\right) &= \mathbb{E}_{\boldsymbol{x}, \left\{\boldsymbol{y}^{(i)}\right\}_{i=1}^{G} \sim \pi_{\theta_{\text{old}}}(\cdot|\boldsymbol{x})}\left\{\left(\hat{A}^{(i)} - \hat{A}_\theta^{(i)}\right)^2 - \beta\,\mathbb{D}_{\text{KL}}\left[\pi_\theta||\pi_{\text{ref}}\right]\right\}, \\
\hat{A}_\theta^{(i)} &= \frac{\rho\left(\boldsymbol{x}, \boldsymbol{y}^{(i)}\right) - \mu_\rho}{\sigma_\rho}, \quad \rho\left(\boldsymbol{x}, \boldsymbol{y}\right) = \log\frac{\pi_\theta\left(\boldsymbol{y}|\boldsymbol{x}\right)}{\pi_{\theta_{\text{old}}}\left(\boldsymbol{y}|\boldsymbol{x}\right)},
\end{aligned} \tag{18}$$

where $\mu_\rho, \sigma_\rho$ are the mean and standard deviation of $\left\{\rho\left(\boldsymbol{x}, \boldsymbol{y}^{(i)}\right)\right\}_{i=1}^{G}$, respectively. For simplicity, we omit the KL divergence between the policy and the reference model.

**Discussion.** Motivated by Group-Relative Policy Optimization (GRPO), the strength of Reg-GRPO lies in its regression-based approach to advantage estimation, unlike other methods that implicitly derive policy updates from preference probabilities. By directly regressing the group-normalized target, Reg-GRPO leads to more precise updates, as the model is not just learning which response is better, but also how much better it is, according to the advantage.

In relation to Direct Preference Optimization (DPO), Reg-GRPO offers a different perspective on leveraging preference data. While DPO learns which response is preferred, Reg-GRPO attempts to capture a finer-grained signal about the degree of preference by directly regressing the advantages. This could be particularly beneficial in scenarios where the difference in quality between preferred and dispreferred responses varies significantly. Furthermore, the group-wise normalization inherent in GRPO, and carried into Reg-GRPO, can offer robustness when dealing with diverse and potentially inconsistently scaled preference data, which may require more careful handling in a pairwise DPO setup.

In contrast to REBEL [45], one of the novel regression-based reinforcement-fine-tuning methods, which regresses the unnormalized pairwise reward gap differences between sampled outputs, our proposed Reg-GRPO framework directly learns to predict the group-based normalized advantage. This shift from pairwise to group-level regression is a design choice that addresses the high variance typically observed in the outputs of video LLMs. By normalizing log-probability ratios within each group, Reg-GRPO mitigates scale discrepancies across batches and enhances the optimization during training. As a result, Reg-GRPO offers a more scalable and effective learning paradigm for fine-tuning language models using preference-based feedback.

# B Reward functions

To compute verifiable rewards, we follow existing works [54, 56, 55] that fine-tune VideoLLMs with GRPO.

**Format reward.** Following existing GRPO-based works [54, 56, 55], we employ a format reward to ensure that the model generates outputs in the desired format. For example, the model is trained to output the thought process with `<think>...</think>` followed by the answer with `<answer>...</answer>`. We use regular expressions to verify whether the outputs satisfy the specified format. The format reward $R_{\text{format}}$ is applied to all tasks:

$$R_{\text{format}} = \begin{cases} 0, & \text{if output does not match the format,} \\ 1, & \text{if output matches the format.} \end{cases} \tag{19}$$

**Accuracy reward.** For tasks such as question answering, we employ an accuracy reward, which is formulated as:

$$R_{\text{acc}} = \begin{cases} 0, & \text{if } \hat{a} \neq a \\ 1, & \text{if } \hat{a} = a, \end{cases} \tag{20}$$

where $a$ is the ground-truth answer and $\hat{a}$ is the model prediction, which is extracted from the regular expressions with `<answer>...</answer>`.

**IoU reward for temporal perception.** We utilize an IoU reward to assess the model's ability to identify the temporal segment described by the input query and localize the target within the video. The IoU reward is defined as:

$$R_{\text{IoU}} = \frac{|\mathcal{P} \cap \mathcal{Q}|}{|\mathcal{P} \cup \mathcal{Q}|}, \tag{21}$$

where $\mathcal{P}$ and $\mathcal{Q}$ are the model prediction set and ground-truth set, respectively. For the temporal grounding task, $\mathcal{P}$ and $\mathcal{Q}$ are defined as the timestamps of events within the video.

## C Detailed Experimental Settings

### C.1 Implementation Details

We implement our code using the PyTorch library [79]. We also adopt the Hugging Face Transformers library [80] and the TRL library [81] to post-train Video Large Language Models (VideoLLMs). For inference and rollout, we use vLLM [82]. For all the experiments, we fine-tune only a large language model while keeping the visual encoder frozen. We use Qwen2.5-VL [7] and Qwen2-VL [5] as our base VideoLLMs. We use NVIDIA A100 GPUs for 3B models and NVIDIA H200 GPUs for 7B models. In addition, we use LLM-based tools for implementation and for correcting grammatical errors in the writing.

For the SEED-Bench-R1 dataset, we apply a KL-divergence regularizer between the model $\pi_\theta$ and the reference model $\pi_{\text{ref}}$ with coefficient 0.1, following prior GRPO works [12, 56]. We use Qwen2.5-VL as the default base VideoLLM, and use Qwen2.5-VL-3B for all analyses. We set the number of generations in the group as 8 for all the settings. For DeepVideo-R1, we maintain a reward history using the most recent $W = 100$ samples, and GRPO does not adopt safeguards based on our empirical study. To train the model on the SEED-Bench-R1 dataset, we limit the maximum number of sampled frames per input video to 16 with a frame resolution of $252 \times 252$, and then append the frame indicating the current observation as an additional image input, following SEED-Bench-R1 [56]. To train the model using NExTGQA, we follow the experimental setups in VideoChat-R1 [55].

### C.2 Evaluation Metrics

**Accuracy.** The accuracy metric measures the ratio of correct predictions that match the ground-truth answers for given questions, which is as follows:

$$\text{Acc} = \frac{1}{N} \sum_{i=1}^{N} \mathbb{1}\left(\hat{a}_i = a_i\right), \tag{22}$$

where $N$ is the number of samples, $\hat{a}_i$ is the prediction, and $a_i$ is the ground-truth answer.

**mIoU.** The mIoU (*i.e.*, mean Intersection over Union) metric calculates the average IoU over all samples, where IoU represents the similarity between the predicted and ground-truth timestamps,

which can be formulated as:

$$\text{mIoU} = \frac{1}{N}\sum_{i=1}^{N}\text{IoU}_i = \frac{1}{N}\sum_{i=1}^{N}\frac{|p_i \cap q_i|}{|p_i \cup q_i|} = \frac{1}{N}\sum_{i=1}^{N}\frac{|\,(s_i^p, e_i^p) \cap (s_i^q, e_i^q)\,|}{|\,(s_i^p, e_i^p) \cup (s_i^q, e_i^q)\,|}, \tag{23}$$

where $N$ is the number of samples, $p_i = (s_i^p, e_i^p)$ is the prediction, $q_i = (s_i^q, e_i^q)$ is the ground truth, and $s_i^p, e_i^p, s_i^q, e_i^q$ denote the start and end timestamps of the prediction and ground truth, respectively.

**R@$m$.** [35] proposed "R@$n$, IoU $= m$" metric for the temporal grounding tasks that measures the percentage of queries where at least one of the top-$n$ predictions has an IoU higher than $m$ with the ground-truth. Following [55], we adopt R@$m$ as a top-1 variant of "R@$n$, IoU $= m$", defined as:

$$\text{"R@}n,\ \text{IoU} = m\text{"} = \frac{1}{N}\sum_{i=1}^{N} \mathbb{1}\left(\text{IoU}_i^j \geq m, \ \exists j \in \{1, 2, \ldots, n\}\right),$$

$$\text{R@}m = \text{"R@}1,\ \text{IoU} = m\text{"} = \frac{1}{N}\sum_{i=1}^{N} \mathbb{1}\left(\text{IoU}_i \geq m\right), \quad \text{where } \text{IoU}_i = \text{IoU}_i^1, \tag{24}$$

where $N$ is the number of samples, $m$ is the IoU threshold, and $\text{IoU}_i^j$ is the IoU between the $j$-th prediction (ranked among the top-$n$ predictions) and the ground truth.

## D  RL Baselines

In this section, we describe the baseline methods used to compare against Reg-GRPO in Table 5 of the main paper.

**DPO [42]** aligns model outputs with human preferences using pairwise comparisons. For DPO, we sample model outputs from a fixed reference model.

**Online DPO [42]** also adopts direct preference optimization to learn the model. Compared to standard DPO, it samples outputs from the old policy model, which evolves during training, following GRPO [12].

**REINFORCE [75–77]** is a classic policy-gradient algorithm that updates the model using the reward-weighted log-likelihood of the outputs. Generally, it also samples the outputs from the old policy model $\pi_{\theta_{\text{old}}}$.

**REINFORCE Leave-One-Out (RLOO) [78]** is designed to reduce the variance of gradient estimates in REINFORCE. Instead of using the reward directly as a weight, it uses a Monte Carlo estimate to compute a baseline and subtracts it from the reward when forming the weight. Same as REINFORCE, it samples the outputs from the old policy model $\pi_{\theta_{\text{old}}}$.

**REBEL [45]** directly regresses the pairwise reward gap, which motivates us to apply the regression-based fine-tuning methods. Different from our work that directly predicts the group-normalized advantage, it regresses the unnormalized pairwise reward gap.

**Reward-Regression (Eq. (5))** is our baseline that directly regresses the reward by approximating $Z(x)$ with Monte-Carlo sampling. Since $Z(x)$ is not accurate and relying solely on the reward introduces high variance, it performs worse than our Reg-GRPO.

## E  Datasets

**SEED-Bench-R1 [56]** is a dataset designed to evaluate the effectiveness of post-training methods in the context of video understanding capabilities of MLLMs. Specifically, the dataset incorporates Epic-Kitchens [83] and Ego4D [84] as videos and EgoPlan-Bench [85] and EgoPlan-Bench2 [86] as benchmark sources to construct a hierarchical validation structure, enabling evaluation across diverse real-world scenarios.

**LongVideoBench [25]** contains 3,763 videos and 6,678 QA pairs, where videos are diverse in domain (*e.g.*, Life, Movie), task (*e.g.*, scene-referred event, object before/after text), and duration. In particular, the video durations are divided into four progressive groups, *(8s, 15s], (15s, 60s], (180s,*

*600s], (900s, 3600s]*, with an overall average of 100s, facilitating the assessment of the model's understanding of long-context interleaved multimodal inputs.

**VSI-Bench** [87] is a dataset proposed to evaluate the visual-spatial intelligence capabilities of MLLMs, comprises over 5,000 question-answer pairs and 288 real videos. Specifically, eight tasks (object count, relative distance, relative direction, route plan, object size, room size, absolute distance, appearance order) of three types (configurational, measurement estimation, spatiotemporal) are defined within the dataset.

**Video-MMMU** [88] consists of 300 expert-level videos and 900 questions, targeting the evaluation of the knowledge acquisition capabilities in MLLMs. Inspired by the human process of acquiring knowledge to solve challenging problems, the questions in the dataset are human-annotated across six disciplines (Art, Business, Science, Medicine, Humanities, Engineering) and aligned with three stages: Perception, Comprehension, and Adaptation.

**MMVU** [89] comprises 3,000 expert-annotated question-answer pairs and 1,529 specialized-domain videos covering 27 subjects across 4 key disciplines (Science, Healthcare, Humanities & Social sciences, Engineering), and aims to evaluate the expert-level, knowledge-intensive video understanding abilities of MLLMs. Following [54], we report the performance on multiple-choice QA.

**MVBench** [27] serves as a benchmark to assess temporal comprehension capabilities of MLLMs, and consists of 20 challenging video understanding tasks that require reasoning beyond a single frame. In particular, the dataset is built upon videos sourced from various benchmarks, enabling the evaluation of MLLMs' general ability for open-world temporal understanding. Since each task contains 200 question-answer pairs, we conduct evaluation on a total of 4,000 question-answer pairs.

**TempCompass** [90] is designed for evaluating the temporal perception ability of MLLM, based on 5 basic temporal aspects (Action, Speed, Direction, Attribute change, Event order) and 10 fine-grained sub-aspects (*e.g.*, relative speed, camera direction, combined change). We report results on overall performance, including all four tasks (Multi-choice QA, Yes/No QA, Caption matching, Caption generation), for comparison with prior work [54].

**Video-MME** [26] is a dataset for evaluating the general video understanding capabilities of MLLMs, consisting of 900 videos and 2,700 question-answer pairs, where the videos are constructed with variation in both type and temporal duration. Specifically, the dataset covers 6 key domains and 30 sub-class video types, and each video is categorized as short (< 2 mins), medium (4-15 minutes), or long (30-60 minutes) depending on its duration length. We report the average performance across all temporal duration splits, without using subtitles.

**NExTGQA** [37] is a temporal grounding QA dataset consisting of 5,417 videos, 43,043 QA pairs, and 10,531 timestamp labels. As temporal segment annotations are available only in the validation and test splits, we use the validation and test splits for the model training and model evaluation, respectively.

## F  Broader Impacts and Limitations

### F.1  Broader Impacts

We propose a video large language model named DeepVideo-R1, trained with Regressive GRPO (Reg-GRPO) and difficulty-aware data augmentation. DeepVideo-R1 is broadly applicable to complex video reasoning tasks. We believe that DeepVideo-R1 itself does not introduce new negative impacts. However, as the model is based on pretrained language and vision models, it may generate biased outputs related to race, religion, culture, and gender, which could lead to misuse. In addition, training VideoLLMs may result in $CO_2$ emissions, which contribute to global warming.

### F.2  Limitations

Our DeepVideo-R1 is built upon a large-scale pretrained video large language model and fine-tuned on video reasoning datasets to leverathe the rich world knowledge embedded in the pretrained models. However, it remains unclear whether there is any overlap between pertaining content and downstream evaluation benchmarks. This uncertainty may introduce a risk of implicit data leakage. Furthermore, since DeepVideo-R1 is based on VideoLLM, it has following limitations: high computational and

memory requirements. As DeepVideo-R1 is fine-tuned on top of such models, it may inevitably inherit these challenges.

