# OpenReview forum: "DeepVideo-R1: Video Reinforcement Fine-Tuning via Difficulty-aware Regressive GRPO"
_NeurIPS.cc/2025/Conference — NeurIPS 2025 poster_

### Official Review · Reviewer_s81Z · 2025-07-02

**Clarity:** 3
**Significance:** 3
**Originality:** 3
**Rating:** 5
**Confidence:** 4

**Summary:**

DeepVideo-R1 enhances video LLMs by addressing two major limitations of GRPO, reliance on heuristic safeguards and the vanishing advantage problem. It introduces Reg-GRPO, it directly predicts advantage values, eliminating problematic safeguards like clipping and min functions. Extensive experiments demonstrate that DeepVideo-R1 significantly improves performance across multiple challenging video reasoning benchmarks, outperforming recent SOTA VideoLLMs such as Qwen2.5-VL and Intern3-VL.

**Questions:**

1. Predicting advantage values may introduce additional training instability if the estimation becomes noisy,

2. Could the computational overhead from difficulty-aware augmentation can be quantified and compared?

3. How robust is the Reg-GRPO to noisy or sparse reward signals in real-world video reasoning tasks?

**Ethical Concerns:**

["NO or VERY MINOR ethics concerns only"]

**Final Justification:**

The author's response resolves my most of concerns. DeepVideo‑R1 with Reg‑GRPO and difficulty‑aware augmentation outperforms baselines across diverse benchmarks with no added inference cost, stable training, and robustness, demonstrating effectiveness and generalizability. I update my score to accept

**Limitations:**

yes

**Paper Formatting Concerns:**

No concerns

**Quality:**

3

**Strengths And Weaknesses:**

Strengths:

- The writing and figures are easy to follow.
- This paper introduces a novel Reg-GRPO approach, effectively addressing vanishing advantage problem and reliance on heuristic safeguards.
- Implements a difficulty-aware data augmentation strategy, effectively maintaining diverse training signals and enhancing the generalizability of video reasoning.
- Demonstrates substantial improvements in performance on complex video reasoning tasks, outperforming SOTA

Weakness :

- Diversifying the VideoLLM baselines and analysis would strengthen the paper, one good candidate would be Video-R1 or VideoLLM with GRPO variants. https://arxiv.org/abs/2503.21776, https://arxiv.org/abs/2506.01908v1 etc
- Potential computational overhead from dynamically adjusting data difficulty during training. I suspect significant computational resources and training time for this approach.
- The generalizability of the proposed augmentation strategy to extremely diverse or real-world scenarios beyond the evaluated benchmarks is not fully explored.

---

> ### Author Rebuttal · Authors · 2025-07-31
>
> We sincerely thank the reviewer for the positive and constructive comments. We are encouraged that **the reviewer recognized multiple strengths** in our paper, including:
>
> (i) **clear writing**
>
> **(ii) The novel Reg-GRPO approach**
>
> **(iii) Effective difficulty-aware data augmentation**
>
> **(iv) Strong empirical results, with DeepVideo-R1 outperforming prior state-of-the-art models on complex video reasoning benchmarks.**
>
> We have tried our best to address the reviewers’ questions and concerns within the available time. We believe that incorporating this constructive feedback significantly enhances the quality of the paper. Below, we provide the response to the given comments:
>
> ---
>
> **Q1. Performance on other video baselines. [Weakness 1]**
>
> **A1.** We have **already demonstrated the effectiveness of our DeepVideo-R1 compared to other GRPO-based VideoLLM such as TimeZero and VideoChat-R1** on both **Charades-STA and ActivityNet-Captions** in Table 10 of the supplement. The table shows that our DeepVideo-R1 achieves the best performance compared to these baselines across both benchmarks. These results highlight the effectiveness of our proposed Reg-GRPO and difficulty-aware data augmentation approaches.
>
> Furthermore, we measured the performance of our DeepVideo-R1 on diverse video understanding reasoning benchmarks such as VideoMMMU, MMVU (mc), MVBench, TempCompass, and Video-MME as reported in Table 9 of the supplement. As suggested, we compare our DeepVideo-R1 with Video-R1 on these benchmarks. We use training data of Video-R1 and use Qwen2.5-VL-3B as a base model. The experimental results in the table below indicate that our DeepVideo-R1 shows better performance on 4 out of 5 benchmark datasets compared to Video-R1, validating the generalizability and robustness of DeepVideo-R1 across diverse video-language reasoning tasks.
>
> |  | Video-MMMU | MMVU (mc) | MVBench | TempCompass | Video-MME (w/o sub) |
> | --- | --- | --- | --- | --- | --- |
> | Video-R1 | 38.9 | 56.5 | 49.1 | **63.4** | 49.7 |
> | **DeepVideo-R1 (ours)** | **40.7** | **59.0** | **49.6** | 63.1 | **51.1** |
>
> ---
>
> **Q2. Computational costs of difficulty-aware data augmentation. [Weakness 2, Question 2]**
>
> **A2.** Good question. As suggested, we compare the computational cost of GRPO, DeepVideo-R1 without difficulty-aware data augmentation, and DeepVideo-R1 by measuring the average training and inference time per video across three different benchmarks (SBR-L1, SBR-L2, SBR-L3) in the table below. We use Qwen2.5-VL-Instruct 3B as a base multimodal large language model and NVIDIA A100 GPUs, following the experimental setting in our supplement.
>
> |  | SBR-L1 | SBR-L2 | SBR-L3 | Inference time (sec/video) | Training time (sec/video) |
> | --- | --- | --- | --- | --- | --- |
> | GRPO | 39.56 | 41.01 | 35.35 | **3.08** | **2.97** |
> | DeepVideo-R1 (w/o D.A. Aug) | 44.20 | 44.15 | 39.52 | **3.08** | **2.97** |
> | DeepVideo-R1 | **48.05** | **51.07** | **43.98** | **3.08** | 4.56 |
>
> The table shows that the inference time remains constant across methods since both Reg-GRPO and D.A. Aug. are training approaches. Notably, DeepVideo-R1 significantly improves the performance of GRPO without increasing inference cost. Even though the difficulty-aware augmentation slightly increases training time, it has a superior performance gain of 6.92 on SBR-L2 compared to the model without difficulty-aware data augmentation, demonstrating the necessity of the augmentation.
>
> ---
>
> **Q3. Generalizability of the difficulty-aware data augmentation. [Weakness 3]**
>
> **A3.** To evaluate the generalizability of difficulty-aware augmentation on different types of video reasoning tasks, we additionally conduct the experiments by comparing the performance of DeepVideo-R1 with and without difficulty-aware augmentation on five video reasoning benchmarks. The experimental results are described in the table below. The table demonstrates that our difficulty-aware data augmentation strategy is broadly applicable and effective beyond SEED-BENCH-R1 datasets.
>
> |  | Video-MMMU | MMVU (mc) | MVBench | TempCompass | Video-MME (w/o sub) |
> | --- | --- | --- | --- | --- | --- |
> | DeepVideo-R1 (w/o D.A. Aug.) | 39.1 | 57.5 | 48.6 | 62.3 | 50.3 |
> | DeepVideo-R1 (with D.A. Aug.) | **40.7** | **59.0** | **49.6** | **63.1** | **51.1** |
>
> ---
>
> **Q4. Training stability of the Reg-GRPO. [Question 1]**
>
> A4. To assess the training stability of Reg-GRPO, we analyze the divergence between $\pi_\theta\left(y|x\right)$ and $\pi_{\theta_{\text{old}}}\left(y|x \right)$. Specifically, we report the average absolute log-ratio $|\log \frac{\pi_\theta \left(y|x \right)}{\pi_{\theta_{\text{old}}}\left(y|x \right)}|$ and the ratio of tokens whose likelihood ratios $\frac{\pi_\theta(y_t | x, y_{<t})}{\pi_{\theta_{\text{old}}}(y_t | x, y_{<t})}$deviate significantly from 1 (e.g., less than 0.5 or greater than 1.5), which indicates instability in updates. The experimental results in the table below show that Reg-GRPO consistently yields smaller deviations than GRPO across all metrics. These findings suggest that Reg-GRPO leads to more stable policy updates, which is crucial for ensuring reliable training. We will include additional plots illustrating these stability metrics over training iterations in the final version to further support this analysis.
>
>
> |  | Avg. $\lvert \log \frac{\pi_\theta} {\pi_{\theta_{\text{old}}}}\rvert$ |
> | --- | --- |
> | GRPO | 0.000861 |
> | Reg-GRPO | 0.000490 |
>
>
> - Avg. token ratio of $\\lvert 1-\frac{\pi_\theta(y_t | x, y_{<t})}{\pi_{\theta_{\text{old}}}(y_t | x, y_{<t})}\\rvert>0.3$
>
>   - GRPO: 0.000972
>   - Reg-GRPO: 0.000435
>
> - Avg. token ratio of $\\lvert 1-\frac{\pi_\theta(y_t | x, y_{<t})}{\pi_{\theta_{\text{old}}}(y_t | x, y_{<t})}\\rvert>0.5$
>
>   - GRPO: 0.000231
>   - Reg-GRPO: 0.000073
>
> - Avg. token ratio of $\\lvert 1-\frac{\pi_\theta(y_t | x, y_{<t})}{\pi_{\theta_{\text{old}}}(y_t | x, y_{<t})}\\rvert>1.0$
>
>   -  GRPO: 0.000036
>   -  Reg-GRPO: 0.000009
>
> ---
>
> **Q5. Robustness of the Reg-GRPO on sparse or noisy reward signals. [Question 3]**
>
> **A5.** Thank you for an insightful and important question. We have already shown the robustness of the Reg-GRPO under **sparse reward signals** where a reward signal is assigned only when the generated output is correct. This setup reflects common conditions in video QA tasks with binary correctness-based supervision. **Table 1 and 4 of the main paper** present results under this sparse reward setting. Notably, **DeepVideo-R1 equipped with Reg-GRPO significantly outperforms GRPO**, highlighting its resilience to sparse reward signals.
>
> To further evaluate the robustness of our proposed Reg-GRPO under **noisy reward signals**, we conduct additional experiments with a newly introduced noise-injected setting. To the best of our knowledge, there has not been an experimental setting using noisy reward signals. So, we newly introduce a noisy-injected reward experimental setup, in which **the reward signal is flipped with 30% probability**: correct outputs are incorrectly assigned a reward of 0, and incorrect outputs receive a reward of 1. Under this setting, we compare GRPO and Reg-GRPO on SEED-Bench-R1 in the table below. To ensure a fair comparison that isolates the effect of the reinforcement learning algorithm, we **exclude difficulty-aware data augmentation** from all models in this experiment.
>
> The experimental results demonstrate that **Reg-GRPO exhibits significantly lower performance degradation than GRPO under noisy rewards**, maintaining relatively stable performance across all datasets. In particular, the performance degradation of Reg-GRPO is only 1.75 on SBR-L3 dataset while GRPO suffers a substantial performance drop of 9.38. This indicates that the regression-based formulation of Reg-GRPO is inherently more robust to noise in reward signals compared to GRPO. We will include this analysis and new experimental setting in the final version of the paper if the paper gets accepted.
>
> |  | SBR-L1  | SBR-L2 | SBR-L3  |
> | --- | --- | --- | --- |
> | GRPO (noise) | 28.29 | 28.17 | 25.97 |
> | GRPO (org) | 39.56 | 41.01 | 35.35 |
> | GRPO (noise - org) | -11.27 | - 12.84 | -9.38 |
> | Reg-GRPO (noise) | 38.19  | 38.29 | 37.77 |
> | Reg-GRPO (org) | 44.20  | 44.15 | 39.52 |
> | Reg-GRPO (noise - org) | -6.01 | -5.86 | -1.75 |

---

> > ### Comment · Reviewer_s81Z · 2025-08-06
> >
> > Thank you for clarifying this. It addresses most of my concerns, I'll raise my score

---

### Official Review · Reviewer_4urM · 2025-07-03

**Clarity:** 3
**Significance:** 2
**Originality:** 3
**Rating:** 4
**Confidence:** 4

**Summary:**

This work rewrites the GRPO loss to mitigate potential gradient suppression and employs difficulty-aware augmentation to address the possible vanishing advantage issue. Based on these two components, the authors propose DeepVideo-R1, which significantly enhances MLLMs' video understanding capabilities through reinforcement learning. The experimental results validate the effectiveness of the algorithm.

**Questions:**

1. There are instances of unclear notation in the paper. Some symbols in Section 3 lack definitions, and $\epsilon$ appears to have different meanings in Equations 2 and 3, while $\pi^\star_{\theta_{old}}$ in Equation 8 should likely be $\pi_{\theta_{old}}$.
2. Minor theoretical issues:

    a) In Equation 8, there **seems** to be no $\epsilon$. This proof is overly brief in the Appendix, while the previous content derived from DPO occupies excessive length. A further elaboration on this proof would be beneficial.

    b) Line 149 should be '*are mean and standard deviation of $log ρ^\star$, respectively*'.

    c) Is Equation 5  a necessary condition or an equivalent condition for  $\pi^\star$ to be an optimal policy? If it’s a necessary condition, how to confirm that the update direction of Equation 9 is correct? Further clarification may be needed.

3. Aside from adding noise for data augmentation related to videos, other components do not seem specific to video tasks. Can Reg-GRPO be directly applied to textual tasks? Or what implicit connections exist between it and video tasks?

4. A more extensive evaluation on various video benchmarks will provide a comprehensive assessment of model capabilities.

5. The methodology regarding data usage seems unclear. Are SEED-Bench-R1 and NextGQA datasets mixed together for RFT or used separately? If they are used separately, can they also be combined?

6. This paper claims that the clipping operation of GRPO may lead to zero gradients, while Reg-GRPO improves upon this point. It is suggested that a comparison between GRPO and Reg-GRPO regarding distances between $\pi_\theta$ and $\pi_{\theta_{old}}$ during training would better validate this claim.

**Ethical Concerns:**

["NO or VERY MINOR ethics concerns only"]

**Final Justification:**

My concerns are mainly resolved throughout the rebuttal. I think this work is valuable. Please don’t forget to revise the faults in theoretical analysis and make clear statements between Reg-GRPO and video understanding.

**Limitations:**

yes

**Paper Formatting Concerns:**

See Questions part

**Quality:**

3

**Strengths And Weaknesses:**

Strengths:
It is valuable to rewrite the GRPO loss term using the closed-form solution from DPO to replace the Clip operation that could lead to zero gradients. Ultimately, the loss aligns the generation probabilities with the advantages in a reasonable manner. The difficulty-aware data augmentation is reasonable for constructing a dataset without the vanishing advantage issue. The experiments reflect the efficacy of Reg-GRPO.

Weaknesses:
Reg-GRPO does not specifically target characteristics of video tasks, and there are some perplexing aspects in the theoretical section. The experimental part could be further enriched to improve overall quality. Please refer to the Questions part, and I would be willing to raise my score if the authors address my concerns.

---

> ### Author Rebuttal · Authors · 2025-07-31
>
> We sincerely thank the reviewer for the valuable and constructive review. We are encouraged that the reviewer highlights multiple strengths of our work, including:
>
> **(i) A principled reformulation of GRPO**
>
> **(ii) Effective advantage alignment via Reg-GRPO**
>
> **(iii) Difficulty-aware data augmentation that addresses the key limitation of GRPO**
>
> **(iv) Comprehensive empirical validation**
>
> We believe that incorporating this constructive feedback significantly enhances the quality of the paper. Below, we respond to the reviewer’s insightful questions below:
>
> ---
>
> **Q1. Typos and clarity of notations [Question 1,2(b)].**
>
> **A1.** We appreciate the feedback. We identified the mentioned typographical issues and some notations with a lack of explanation. We will reflect them in the final version if the paper gets accepted.
>
> ---
>
> **Q2. More detailed derivation of reg-grpo. [Question 2(a)]**
>
> **A2.** As requested, we specifically clarify why $\epsilon$ is needed in Eq. (8) of the supplement as below.
>
> Our optimization objective is :
>
> $$
> \max_{\pi} \ \mathbb{E}_{ x,  y\sim\pi\_\theta (\cdot |  x )} \mathcal{R} ( x,  y ) - \epsilon \cdot \mathbb{D}\_{\text{KL}} (\pi\_\theta || \pi\_{\theta\_{\text{old}}} ),
> $$
>
> where $\epsilon$ is a regularization strength.
>
> From Section A.1 of the supplement in the DPO paper [1] and our supplement, we can obtain that the optimal policy for our objective is
>
> ${\pi}\_{\theta}^* ( y |  x  ) = \frac{1}{Z ( x )} \pi\_{\theta_{\text{old}}} ( y |  x  )\exp (\frac{1}{\epsilon} \mathcal{R} ( x,  y  )  ),\ Z(x) = \sum_y \pi\_{\theta\_{\text{old}}} (y|x)\exp  (\frac{1}{\epsilon} \mathcal{R}(x,y)  )$
>
> where $\mathcal{R} (x,y )$ is the reward value given $x$ and $y$.
>
> From this equation, the reward $\mathcal{R} ( {x}, {y} )$ can be formulated under $\pi_{\theta_{\text{old}}} ( {y} |  {x}  )$ and $\pi_\theta ( {y} |  {x}  )$.
>
> By multiplying both sides of Eq. (5) by $Z ( {x} )$, we can get
>
> $Z ( x ) \cdot {\pi}\_{\theta}^* ( y|  x ) =  \pi\_{\theta\_{\text{old}}} ( y |  x  )\exp (\frac{1}{\epsilon} \mathcal{R} ( x,  y )  ),$
>
> $Z ( x )\cdot \frac {{\pi}\_{\theta}^* ( {y} |  {x}  )}{\pi\_{\theta\_{\text{old}}} ( {y} |  {x}  )} =  \exp (\frac{1}{\epsilon} \mathcal{R} ( x,  y  )  ),$
>
> Next, we take the logarithm of both sides and then multiply both sides by $\epsilon$ to obtain the following equation (Eq. (6) of the supplement):
>
> $\mathcal{R} ( {x},  {y}  )  = \epsilon \cdot  (\log Z ( {x}  ) + \log  (\frac{\pi_{\theta}^* ( {y} |  {x}  )}{\pi_{\theta_{\text{old}}} ( {y} |  {x}  )}  ) ),$
>
> The advantage of GRPO is defined as
>
> $A^{(i)} = \frac{\mathcal{R} ( {x},  {y}^{(i)}  )-\mu_r}{\sigma_r}$. (Eq. (7) of the supplement), where $\mu_r, \sigma_r$ are mean and standard deviation of a set of reward values $\\{\mathcal{R} ( {x},  {y}^{(i)}  ) \\}_{i=1}^G$.
>
> Using Eq.(6) of the supplement, we can reformulate the advantage by substituting $\mathcal{R} ( {x},  {y} )$ with $\epsilon \cdot  (\log Z ( {x}  ) + \log  (\frac{\pi_{\theta}^* ( {y} |  {x}  )}{\pi_{\theta_{\text{old}}} ( {y} |  {x}  )}  ) )$. It can be written as
>
> $$
> \hat{A}^{(i)} = \frac{\epsilon \cdot \log Z ( {x}  ) + \epsilon \cdot\log  (\frac{\pi_{\theta}^* ( {y}^{(i)} |  {x}  )}{\pi_{\theta_{\text{old}}} ( {y}^{(i)} |  {x}  )}  ) - \epsilon \cdot \log Z ( {x}  ) - \epsilon \cdot \frac{1}{G} \sum_{i=1}^G\log  (\frac{\pi_{\theta}^* ( {y}^{(i)} |  {x}  )}{\pi_{\theta_{\text{old}}} ( {y}^{(i)} |  {x}  )}  )}{\sigma_\rho}
> $$
>
> $$
> \hat{A}^{(i)} = \frac{ \epsilon \cdot \log  (\frac{\pi\_{\theta}^* ( {y}^{(i)} |  {x}  )}{\pi\_{\theta_{\text{old}}} ( {y}^{(i)} |  {x}  )}  ) - \epsilon \cdot \frac{1}{G} \sum\_{i=1}^G\log  (\frac{\pi\_{\theta}^* ( {y}^{(i)} |  {x}  )}{\pi\_{\theta\_{\text{old}}} ( {y}^{(i)} |  {x}  )}  )}{\sigma\_\rho}
> $$
>
> $$
> = \epsilon \cdot \frac{  \log \rho^* ( x, y^{(i)}  ) - \mu\_\rho^\*}{\sigma\_\rho^\*},\quad \rho^* ( {x},  {y}  ) = \frac{\pi\_{\theta}^* ( {y} |  {x}  )}{\pi\_{\theta\_{\text{old}}} ( {y} |  {x}  )}, \quad  \mu\_\rho^* = \frac{1}{G}\sum\_{i=1}^G\log  (\frac{\pi\_{\theta}^* ( {y}^{(i)} |  {x}  )}{\pi\_{\theta\_{\text{old}}} ( {y}^{(i)} |  {x}  )}  ), \quad \sigma\_\rho^* = \text{Std} (\\{\log \rho^* ( {x},  {y}^{(i)}  ) \\}\_{i=1}^G  )
> $$
>
>
> Thus, we can obtain Eq. (8) and $\epsilon$ is required. It arises naturally from the KL-regularized RL objective and controls the strength of the regularization and scale of the reward. We have updated the supplement to make this derivation more explicit and easier to follow.
>
> ---
>
> **Q3. Is Eq (5) in the supplement a necessary condition or an equivalent condition for** $\pi^*$ **to be an optimal policy? [Questions 2(c)]**
>
> **A3.**  $\pi^*$ is an optimal policy and the condition is **an equivalent (necessary and sufficient) condition** as proven from Section A in the supplement of DPO paper [1].
>
> Based on the result, our derivation is provided in Section A.1 of our supplement.
>
> We provide a sketch of proof here for the convenience.
>
> The policy optimization objective is given as:
>
> $$
> \pi_\theta^* = \text{arg}\max\_\pi \ \mathbb{E}\_{ {x},  {y}\sim\pi (\cdot |  {x} )} \mathcal{R} ( {x},  {y} ) - \epsilon \cdot \mathbb{D}\_{\text{KL}} (\pi\_\theta || \pi\_{\theta\_{\text{old}}} )
> $$
>
> From our supplement and other works, the above problem can be reformulated as
>
> $$
> \arg \min_\pi\ \mathbb{E}\_{ {x}}\left[\mathbb{D}\_{\text{KL}} (\pi_\theta  ( {y} |  {x}  )||\bar{\pi}_\theta  ( {y}| {x}  )  )-\log Z ( {x} ) \right],
> $$
>
> where $\bar{\pi}\_{\theta} ( {y} |  {x}  ) = \frac{1}{Z ( {x} )} \pi\_{\theta\_{\text{old}}} ( {y} |  {x}  )\exp (\frac{1}{\epsilon} \mathcal{R} ( {x},  {y}  )  )$. Since the partition function does not depend on $\pi$, the optimal policy minimizes the KL regularization.
>
> The minimum KL is achieved **if and only if** two distributions are identical, the optimal solution is Eq. (5) as follows
>
> $$
> {\pi}\_{\theta}^* ( {y} |  {x}  ) = \frac{1}{Z ( {x} )} \pi_{\theta_{\text{old}}} ( {y} |  {x}  )\exp (\frac{1}{\epsilon} \mathcal{R} ( {x},  {y}  )  ), \forall  {x},  {y}
> $$
>
> Thus, Eq (5) is a necessary and sufficient (equivalent) condition for $\pi$ to be an optimal policy under our objective. So, the update direction of Eq. (9) in our supplement is correct.
>
> ---
>
> **Q4. What connection exists between Reg-GRPO and video tasks? [Question 3]**
>
> **A4.** Good question.  Reg-GRPO (Regressive Group Relative Policy Optimization) is a generic technique designed to enhance the reasoning capability of LLM-based models, particularly for complex tasks. It is not limited to video LLMs. To show the generalizability of Reg-GRPO, we evaluate it by comparing it with GRPO on multiple complex “mathematical reasoning” benchmarks in the table below.
>
> |  | AMC23 | Minerva | MATH500 | OlympiadBench |
> | --- | --- | --- | --- | --- |
> | GRPO | 47.5 | 32.35 | 80.2 | 38.96 |
> | Reg-GRPO | **60.0** | **34.48** | **80.6** | **41.93** |
>
> The performance gain across these diverse reasoning datasets shows that Reg-GRPO is an effective post-training strategy for enhancing LLM reasoning. This generalizability extends naturally to challenging tasks such as video question answering, where complex temporal and causal reasoning is often required. Thus, Reg-GRPO strengthens the core reasoning ability of LLMs, which is essential for solving video tasks that demand high-level understanding and long-term dependencies.
>
> ---
>
> **Q5. Additional performance comparisons on video benchmarks. [Question 4]**
>
> **A5.** To further validate the generalizability of our method, we compare **DeepVideo-R1** against **Video-R1** using *identical pretraining data* and **Qwen2.5-VL-3B** as the base model. Results on five video-language reasoning datasets are shown below:
>
> |  | Video-MMMU | MMVU (mc) | MVBench | TempCompass | Video-MME (w/o sub) |
> | --- | --- | --- | --- | --- | --- |
> | Video-R1 | 38.9 | 56.5 | 49.1 | **63.4** | 49.7 |
> | **DeepVideo-R1 (ours)** | **40.7** | **59.0** | **49.6** | 63.1 | **51.1** |
>
> DeepVideo-R1 outperforms Video-R1 on 4 out of 5 datasets, highlighting the robustness and generalizability of our approach across diverse video-language reasoning tasks.
>
> ---
>
> **Q6. Are the SEED-Bench-R1 and NextGQA datasets combined or used separately? [Question 5]**
>
> **A6.** As noted in the supplement, SEED-Bench-R1 and NextGQA datasets are used separately for reinforcement fine-tuning. In response to the suggestion, we conducted an additional experiment using **joint fine-tuning** on the combined dataset (SEED-Bench-R1 + NextGQA). The results are summarized below:
>
> |  | SBR-L1 | SBR-L2 | SBR-L3 | NexTGQA (mIoU) | NexTGQA (ACC@QA) |
> | --- | --- | --- | --- | --- | --- |
> | SEED-Bench-R1 | 48.05 | 51.07 | 43.98 | - | - |
> | NextGQA | - | - | - | 36.8 | **72.5** |
> | SEED-Bench-R1 + NextGQA | **49.34** | **51.89** | **44.38** | **37.1** | **72.5** |
>
> The results indicate that joint fine-tuning on both datasets slightly improves performance on SEED-Bench-R1 while maintaining comparable performance on NextGQA.
>
> ---
>
> **Q7. Comparison of Reg-GRPO with GRPO regarding the gap between** $\pi_\theta$ and $\pi_\text{old}$. **[Question 6]**
>
> **A7.** Good point. As suggested, we analyze the divergence between $\pi_\theta (y|x )$ and $\pi_{\theta_{\text{old}}} (y|x  )$. Specifically, we report the average absolute log-ratio $|\log \frac{\pi_\theta  (y|x  )}{\pi_{\theta_{\text{old}}} (y|x  )}|$, which indicates instability in updates. The experimental results in the table below show that Reg-GRPO consistently yields smaller deviations than GRPO. It suggests that Reg-GRPO leads to more stable policy updates. We will include additional plots illustrating these stability metrics over training iterations in the final version to further support this analysis.
>
> |  | Avg. $\| \log \frac{\pi_\theta} {\pi_{\theta_{\text{old}}}}\|$ |
> | --- | --- |
> | GRPO | 0.000861 |
> | Reg-GRPO (Ours) | 0.000490 |
>
> [1] Rafailov, Rafael, et al. "Direct preference optimization: Your language model is secretly a reward model." NeurIPS 2023.

---

> > ### Comment · Reviewer_4urM · 2025-08-04
> >
> > Thanks for the concise response, which has mainly addressed my concern.
> > I still have two more questions.
> >
> > 1. why do you simply replace $\sigma_r$ with $\sigma_\rho$ in the proof of Eq(8)?
> >
> > from:
> > $$ A^{(i)}=\frac{\mathcal{R}\left(x, y^{(i)}\right)-\mu_{r}}{\sigma_{r}} $$
> > to:
> > $$ \hat{A}^{(i)} = \frac{\epsilon \cdot \log Z ( {x} ) + \epsilon \cdot\log (\frac{\pi_{\theta}^* ( {y}^{(i)} | {x} )}{\pi_{\theta_{\text{old}}} ( {y}^{(i)} | {x} )} ) - \epsilon \cdot \log Z ( {x} ) - \epsilon \cdot \frac{1}{G} \sum_{i=1}^G\log (\frac{\pi_{\theta}^* ( {y}^{(i)} | {x} )}{\pi_{\theta_{\text{old}}} ( {y}^{(i)} | {x} )} )}{\sigma_\rho} $$
> >
> > 2. What data are used to finetune for the evaluation on the “mathematical reasoning” benchmarks? Why are the enhancements mainly reflected in the 'ACM23'?

---

> ### Author Response · Authors · 2025-08-05
>
> We sincerely thank the reviewer for their continued engagement and are pleased that most concerns have been addressed. Below are our responses to the follow-up questions:
>
> ---
>
> FQ1. Why do you replace $\sigma_r$ with $\sigma_\rho$ in Eq(8)?
>
> We appreciate the reviewer’s feedback. This is a typo and $\epsilon$ was omitted. Sorry for the confusion. It should be $\epsilon \cdot\sigma_\rho = \sigma_r$.
>
> In our experiments, we used the above form as L811-L816 of “grpo_trainer.py” in the supplementary materials. Furthermore, as noted in L139 of the supplement PDF, we used $\epsilon=1$ in the experiments. Thus, our experiments have no discrepancy with the equation above. We will update the manuscript in the final version.
>
> ---
>
> FQ2. What data are used to finetune for the evaluation on the “mathematical reasoning” benchmarks? Why are the enhancements mainly reflected in the 'AMC23'?
>
> Good question. For reinforcement fine-tuning the model, we follow the work [1] and use Math3to5, which consists of math problems at 3-5 difficulty levels of problems in MATH dataset [2]. We believe that the significant performance improvement on **AMC23** arises from the **strong generalization capability** of our proposed Reg-GRPO method. Notably, **AMC23** and **OlympiadBench** are *out-of-distribution* benchmarks that contain high-level, competition-style math problems, whereas **MATH-500** and **Minerva** are more *in-distribution*, aligned with the training distribution of Math3to5.
>
> As shown in Table A4, Reg-GRPO yields the largest gains on the more challenging, out-of-distribution datasets (AMC23 and OlympiadBench). This suggests that Reg-GRPO improves the model's robustness and ability to generalize beyond the training distribution.
>
> [1] Liu, Ziru, et al. "GHPO: Adaptive Guidance for Stable and Efficient LLM Reinforcement Learning." arXiv:2507.10628 (2025).
>
> [2]  Hendrycks, Dan, et al. "Measuring mathematical problem solving with the math dataset." NeurIPS (2021).

---

> > ### Comment · Reviewer_4urM · 2025-08-05
> >
> > Thanks for the newest response.  I’ll raise the score to borderline accept.

---

### Official Review · Reviewer_ZcMY · 2025-07-03

**Clarity:** 3
**Significance:** 3
**Originality:** 3
**Rating:** 5
**Confidence:** 4

**Summary:**

The paper introduces DeepVideo-R1, a novel approach to enhancing the reasoning capabilities of VideoLLMs. The authors identify two key issues in GRPO used for training VideoLLMs: the reliance on heuristic stabilizers and the vanishing advantage problem. To address these, they propose a solution combining Regressive GRPO and difficulty-aware data augmentation. Through experiments across various video reasoning benchmarks, the model demonstrated significant performance improvements over existing models like Qwen2.5-VL and Intern3-VL. The results show that the DeepVideo-R1 model generalizes better to both in-distribution and out-of-distribution tasks, making it more robust than other VideoLLMs.

**Questions:**

I am curious about the regression objective without any clip function inside. Since the original PPO or GRPO will try to truncate the loss when there are some significant shifts. But the proposed method does not follow this and make any ablation about it. Is the training process stable? There should be some ablation curves to illustrate this.

And predicting advantage seems more like a reweighted imitation algorithm, but there is no related works or experiments.

**Ethical Concerns:**

["NO or VERY MINOR ethics concerns only"]

**Final Justification:**

Resolved: 1) Compute practicality clarified—unchanged inference cost, moderate training overhead with clear gains; (2) Difficulty-aware augmentation improves results across five extra benchmarks; (3) Reg-GRPO stable without clipping (endpoint ablations); (4) Relation to reweighted imitation learning now articulated.

Unresolved: Lack of training curves/variance stats; limited full compute reporting (GPU hours/memory/scaling); no direct comparison to advantage-weighted imitation baselines; limited sensitivity analyses.

Camera-ready asks: Add training dynamics/variance plots, fuller compute/scaling details, a small imitation-baseline comparison, and sensitivity/error analyses.

**Limitations:**

As follows.

**Quality:**

3

**Strengths And Weaknesses:**

Strengths

1. The work presents a well-structured and thorough explanation of the novel methods. The authors validate their approach with strong experimental results across multiple challenging benchmarks.

2. The paper is clearly written and the methodology is well-explained, making it accessible to readers with a background in reinforcement learning and multimodal AI.

3. The combination of Reg-GRPO with difficulty-aware augmentation is novel, presenting a fresh approach to improving video reasoning in LLMs.

Weakness:

1. **Limited Discussion of Computational Complexity:** While the paper provides extensive experimental results, it could further discuss the computational cost of the proposed methods, especially with large video datasets.

2. **Generalizability of Difficulty-Aware Augmentation:** The difficulty-aware data augmentation strategy is an interesting approach, but it would be beneficial to explore how well this method scales across different types of video reasoning tasks beyond those tested in the paper.

---

> ### Author Rebuttal · Authors · 2025-07-31
>
> We sincerely thank the reviewer for the positive and constructive review of our paper. We are encouraged that the reviewer recognizes multiple strengths in our paper, including:
>
> **(i) novel and fresh approaches**
>
> **(ii) well-structured and thorough explanation**
>
> **(iii) strong experimental results across multiple challenging benchmarks**
>
> We believe that incorporating this constructive feedback significantly enhances the quality of the paper. Below, we provide the response to the given comments in detail:
>
> ---
>
> **Q1. Discussion of computational complexity. [Weakness 1]**
>
> **A1.** We appreciate the reviewer’s suggestion to study the computational cost of our proposed method. To explore the computational cost, we report the average training and inference time per video across three different benchmarks (SBR-L1, SBR-L2, SBR-L3) in the table below. We use Qwen2.5-VL-Instruct 3B as a base multimodal large language model and NVIDIA A100 GPUs following the experimental setting in our supplement.
>
> |  | SBR-L1 | SBR-L2 | SBR-L3 | Inference time (sec/video) | Training time (sec/video) |
> | --- | --- | --- | --- | --- | --- |
> | GRPO | 39.56 | 41.01 | 35.35 | **3.08** | **2.97** |
> | DeepVideo-R1 (w/o D.A. Aug) | 44.20 | 44.15 | 39.52 | **3.08** | **2.97** |
> | DeepVideo-R1 | **48.05** | **51.07** | **43.98** | **3.08** | 4.56 |
>
> The table shows that the inference time remains constant across methods since our approach focuses on training. Notably, DeepVideo-R1 significantly improves the performance of GRPO without increasing inference cost. Even though the difficulty-aware augmentation slightly increases training time, it has a superior performance gain of 6.92 on SBR-L2 compared to the model without difficulty-aware data augmentation. This demonstrates the effectiveness of the augmentation by addressing the vanishing advantage problem.
>
> ---
>
> **Q2. Generalizability of difficulty-aware augmentation. [Weakness 2]**
>
> **A2.** To evaluate the generalizability of difficulty-aware augmentation on different types of video reasoning tasks, we additionally conduct the experiments by comparing the performance of DeepVideo-R1 with and without difficulty-aware augmentation on five video reasoning benchmarks. The experimental results are described in the table below. The table demonstrates that our difficulty-aware data augmentation strategy is broadly applicable and effective beyond SEED-BENCH-R1 datasets.
>
> |  | Video-MMMU | MMVU (mc) | MVBench | TempCompass | Video-MME (w/o sub) |
> | --- | --- | --- | --- | --- | --- |
> | DeepVideo-R1 (w/o D.A. Aug.) | 39.1 | 57.5 | 48.6 | 62.3 | 50.3 |
> | DeepVideo-R1 (with D.A. Aug.) | **40.7** | **59.0** | **49.6** | **63.1** | **51.1** |
>
> ---
>
> **Q3. Ablation of Reg-GRPO with/without clipping function. [Question 1]**
>
> **A3.** Following the reviewer’s suggestion, we conduct additional experiments comparing DeepVideo-r1 trained with and without applying the clipping mechanism to investigate the impact of the clipping function on training stability. As NeurIPS policy does not allow figure uploads during the rebuttal phase, we provide only final performance instead of providing training curves. The experimental results are presented in the table below. The results show that the performance is comparable with and without clipping functions across all datasets, indicating that Reg-GRPO remains stable and effective without the clipping function. We will include the full training curve and analysis in our final version if the paper gets accepted.
>
> |  | SBR-L1 | SBR-L2 | SBR-L3 |
> | --- | --- | --- | --- |
> | DeepVideo-R1 (with clip. function) | 47.87 | **51.20** | 43.64 |
> | DeepVideo-R1 (without clip. function) (ours) | **48.05** | 51.07 | **43.98** |
>
> ---
>
> **Q4. Relation to reweighted imitation learning algorithm. [Question 2]**
>
> **A4.** Thank the reviewer for the insightful comment regarding the reweighted imitation learning and our Regressive GRPO (Reg-GRPO). As the reviewer pointed out, our Reg-GRPO is related to prior imitation learning methods [1,2,3], which are intuited by a KL-regularized optimization problem. However, while they focus on learning a reward function relying on **”expert demonstrations”** to replicate expert actions, our Reg-GRPO does **NOT** require any expert trajectories or access to an expert policy. Instead, Reg-GRPO directly optimizes a policy by regressing onto the estimated advantages, which are derived from the model’s own rollouts.
>
> We agree that discussing Reg-GRPO with reweighted imitation learning would improve the paper, and we will include a more comprehensive discussion and comparison in the final version. We are also happy to incorporate any additional related works the reviewer may suggest.
>
> [1] Jacq, Alexis, et al. "Learning from a learner." ICML, 2019.
>
> [2] Watson, Joe, Sandy Huang, and Nicolas Heess. "Coherent soft imitation learning." NeurIPS, 2023.
>
> [3] Xu, Haoran, et al. "An Optimal Discriminator Weighted Imitation Perspective for Reinforcement Learning." ICLR, 2025.

---

> > ### Comment · Reviewer_ZcMY · 2025-08-06
> >
> > Thank you for addressing some of my concerns. I will keep my score of 5.

---

### Comment · Area_Chair_1bko · 2025-08-03
**Post-rebuttal Discussion**

The author has provided a rebuttal to respond to your comments. Please take a look on the author response and discuss with authors if necessary.

Thanks,

AC

---

### Note · Authors · 2025-08-15

Dear AC and Reviewers,

We sincerely thank you for the constructive feedback, which has greatly strengthened our paper. We're pleased that all reviewers recognized the novelty and effectiveness of our approach, highlighting the following:

- **Reg-GRPO:** We reformulate the GRPO loss into a new regression objective that aligns the generation probabilities. All three reviewers (Reviewer ZcMY, 4urM, s81Z) recognized our work as a “novel” approach. Our method removes the heuristic safeguards that could lead to suboptimal optimization, and Reviewer 4urM praised that it is “valuable”.
- **Difficulty-aware data augmentation.** We design a new difficulty-aware data augmentation strategy that dynamically adjusts the difficulty of the input problem. All three reviewers (Reviewer ZcMY, 4urM, s81Z) agree on its effectiveness in addressing the *vanishing advantage problem* of GRPO.
- **Strong experimental results of DeepVideo-R1, surpassing SOTA.** Our model significantly improves the reasoning capabilities of video LLMs on complex video reasoning tasks. Reviewers ZcMY and s81Z recognized its strong performance across multiple challenging benchmarks, noting that it outperforms SOTA methods.

We appreciate the opportunity to provide more detailed responses and additional experiments that addressed all reviewer concerns, leading to **positive scores from all reviewers**. Through the rebuttal and author-reviewer discussion, we demonstrate:

- **Effective Generalization** across eight tasks—four video reasoning and four mathematical reasoning—where DeepVideo-R1 outperforms the GRPO baseline (A4, A5 in response to 4urM; A1 in response to s81Z).
- **Superior Robustness & Stability** of Reg-GRPO, consistently yielding smaller policy deviations $|\log \frac{\pi_\theta(y|x)}{\pi_{\theta_{\text{old}}}(y|x)}|$ and delivering significant gains under noisy conditions compared to GRPO (A7 in response to 4urM; A4, A5 in response to s81Z).
- **No Additional Inference Cost**, with DeepVideo-R1 improving GRPO’s performance while maintaining efficiency (A1 in response to ZcMY; A2 in response to s81Z).

We believe that these additional results and discussions will significantly enhance our paper, leading to stronger support. We sincerely thank all the reviewers and Area Chair.

---

### Decision · Program_Chairs · 2025-09-17

**Decision:**

Accept (poster)

**Comment:**

**Summary:** This paper introduces Reg-GRPO, a regularized variant of GRPO that eliminates the need for heuristic clipping operations by reformulating the loss using a closed-form solution inspired by DPO. This mitigates the vanishing-gradient issue and provides a more principled optimization objective. Additionally, the paper proposes a difficulty-aware data augmentation strategy to ensure the model consistently receives effective training signals, thereby improving the generalization ability of Video LLMs on challenging video reasoning tasks. The authors validate their approach on multiple benchmarks, reporting consistent and substantial gains over existing methods.

**Strengths:**

- Reg-GRPO directly addresses the limitations of reliance on safeguards and vanishing advantage in GRPO.

- The difficulty-aware augmentation strategy provides a fresh perspective on dataset construction and signal diversity, and is particularly suitable for video reasoning tasks.

- The empirical results are strong, showing significant improvements across challenging benchmarks and surpassing several strong baselines.

- The paper is clearly written and well-structured, making both the methodology and motivation easy to follow.


**Weaknesses:**

- Computational complexity and scalability: Both Reg-GRPO training and difficulty-aware augmentation may introduce significant overhead, which is not thoroughly quantified or discussed.

- Generalizability of augmentation: While effective on tested benchmarks, the broader applicability to diverse or real-world datasets remains unclear.

Overall, the rebuttal was constructive and effective: it clarified theoretical concerns, acknowledged limitations, and provided convincing arguments about the practicality of the approach. After weighing the discussion, I recommend acceptance.